

# Effect of hydro-climate variation on biofilm dynamic and impact in intertidal environment

Elena Bastianon[1], Julie A. Hope[1], Robert M. Dorrell[1], Daniel R. Parsons[1]

[1] Energy and Environment Institute, University of Hull, Hull, HU6 7RX, United Kingdom

*Correspondence to*: Elena Bastianon (E.Bastianon@hull.ac.uk)

**Abstract.** Shallow tidal environments are very productive ecosystems yet are sensitive to environmental changes and sea level rise. Bio-morphodynamic control of these environments is therefore a crucial consideration; however, the effect of small-scale biological activity on large-scale cohesive sediment dynamic like tidal basins and estuaries is still largely unquantified. This study advances our understanding by assessing the influence of biotic and abiotic factors on biologically

cohesive sediment transport and morphology. An idealised benthic biofilm model is incorporated in a 1D morphodynamic model of tide-dominated channels. By carrying out a sensitivity analysis of the bio-morphodynamic model, i) carpet-like erosion; ii) seasonality; iii) biofilm growth rate; iv) temperature variation; and v) bio-cohesivity of the sediment ($\alpha$); this study allows the effect of a range of environmental and biological conditions on biofilm growth to be investigated, and the feedback on the morphological evolution of the entire intertidal channel. Results reveal that key parameters such as growth

rate and temperature strongly influence the development of biofilm and are key determinants of equilibrium biofilm configuration and development, under a range of disturbance periodicities and intensities. Long-term simulations of intertidal channel development demonstrate that the hydrodynamic disturbances induced by tides play a key role in shaping the morphology of the bed, and the presence of surface biofilm increases the time to reach morphological equilibrium. On the other hand, in locations characterized by low hydrodynamic forces the biofilm grows and stabilizes the bed, inhibiting the

transport of coarse sediment (medium and fine sand). These findings suggest biofilm presence in channel beds results in intertidal channels that have significantly different characteristics in terms of morphology and stratigraphy compared abiotic sediments. It is concluded that inclusion of biocohesion in morphodynamic models is essential to predict and mitigate estuary development and coastal erosion.

## 1. Introduction

Tidal inlets are some of the most sensitive systems to sea-level rise and environmental change. Their morphology is shaped and influenced by tides, waves, river discharge and associated sediment supply of marine and riverine sands and muds (Corenblit et al., 2007; De Haas et al., 2018). The availability of nutrients and sediment from the surrounding area in combination with dynamic environmental conditions, provide a favourable setting for numerous aquatic species, making them one of the most ecologically important environments (Meire et al., 2005). Even though strongly driven by abiotic



processes, biotic processes can determine the geomorphological evolution of intertidal areas (Defew et al., 2002; Malarkey et al., 2015; Parsons et al., 2016; Vignaga et al., 2013). In order to manage these systems and adapt for future changes, there is the need for models that are able to incorporate the role of biocohesion on geomorphology. Those currently available are not yet robust enough to predict, with confidence, very far into the future. Consequently, understanding the interactions between hydrodynamics, sediment erosion and deposition, and biological communities becomes crucial for the sustainable

management of estuaries and intertidal environments.

Biological activity on the seabed is known to have a significant influence on the bed composition and dynamics of cohesive and non-cohesive sediment at both small spatial and temporal scales (Decho, 2000). The presence of benthic microorganisms and the substances that they secrete strongly mediate the physical behaviour and functionality of the depositional system, influencing the structure and behaviour of sedimentary habitats, acting as ecosystem engineers (Paterson, 1997; Paterson et

al., 2018). Microphytobenthos (MPB) is an assemblage of microbial cells, e.g., diatoms, cyanobacteria and heterotrophic bacteria, aggregated within a gel matrix composed of a mixture of lipids, proteins and polysaccharides, known as Extracellular Polymeric Substances (EPS), that form benthic biofilms in intertidal and subtidal areas (Austin et al., 1999; Decho, 2000; Paterson et al., 1994; Tolhurst et al., 2002; Underwood and Paterson, 1993). It has been shown that secreted EPS plays a crucial role in the adhesion/cohesion of the substratum and sediment particles, and it can act as a protective layer

at the bed surface reducing the bed roughness, influencing significantly the erosion and deposition of sediment particles by raising the sediment erosion threshold due to cohesion (Tolhurst et al. 2002, Tolhurst et al. 2006, Tolhurst et al. 2009, Paterson et al., 2018, Hope et al. 2020). This promotes the sedimentation of fine-grained particles and subsequently stimulates biofilm growth (Weerman et al., 2010). Even at low EPS content (Tolhurst et al., 2002) both EPS concentrations (quantity) and EPS components (quality) play important roles on the binding effect on sediment particles; increasing the

critical threshold for erosion, thereby reducing sediment resuspension and bed erosion (Lubarsky et al., 2010; Malarkey et al., 2015; Parsons et al., 2016). This enhanced 'biostabilisation' (Paterson et al., 1989; Tolhurst et al., 2002; Widdows et al., 2000) in turn increase the resistance of bed sediments to erosion and inhibits biofilm displacement influencing the erodibility of the bed sediment (Friend et al. 2003; Amos et al. 2004; Lundkvist et al. 2007), allowing the spatial development of biofilms and stabilization across large geomorphological features (Weerman et al., 2010, Friend et al., 2008). By reducing

the concentration of fine sediment resuspended and consequently the turbidity of the water column, biostabilisation allows more light penetration to the sediment surface, creating a positive feedback to the biofilm community and more growth. Biostabilisation also limits the resuspension of coarse particles that, by moving, could cause abrasion to the biofilm layer and the removal of large sections of biofilm from the bed (Lanuru et al., 2007). Further, the stabilization of the water-sediment interface by benthic biofilm is important for the regulation and bentho-pelagic exchange of carbon, nitrogen and oxygen with

the substrate (Cahoon 1999) and subsequently the transfer of energy and resource to adjacent habitats (Savage et al., 2012). These processes are complicated by the presence of benthic bioturbators that disrupt and graze on MPB, and they can have a high impact on mudflat morphology because they can physically destabilise the bed (e.g. de Deckere et al. 2001, Brückner et al., 2021) and trigger sediment resuspension that is otherwise stabilized by diatoms. Furthermore, the establishment of





biostabilizers might be affected by sediment destabilization and seed predation from bioturbators (Cozzoli et al., 2019). In
turn, bioturbators organically enrich the sediment via biodeposition which can promote the MPB growth (e.g. Andersen et
al., 2010; Donadi et al., 2013); and biostabilizers can modify the hydrodynamics and sediment properties around them
(Brückner et al., 2020), impacting the size and density of bioturbators communities (Walles et al., 2015).

While microbially produced EPS is more abundant in cohesive sediment (muddy bed), studies have shown that EPS
production by bacteria and microphytes can also play a significant role in non-cohesive and mixed sediment substrates by
hindering bedform development and inhibiting erosion (Malarkey et al., 2015, Parsons et al., 2016, Chen et al., 2017, Hope
et al., 2020). The influence of benthic biofilms and EPS on sediment erosion is widely recognized and characterised across
different sedimentary habitats (e.g. Paterson, 1989; MacIntyre et al., 1996; Marani et al., 2010; Malarkey et al., 2016; Hope
et al., 2020; Chen et al., 2021), but few numerical studies account for these processes. The exclusion of biocohesion and
biostabilisation effects makes it difficult for predictive models of sediment stability to be sufficiently accurate. This is
primarily due to the difficulty of simultaneously coupling the physical, biological and biodiversity components. Seasonal
changes in environmental conditions and grazer communities can mediate biofilm grow rate (Underwood and Paterson,
2003; Montani et al., 2003; Zhang et al., 2021; Daggers et al., 2020, Brückner et al., 2021), but interannual changes in key
biota, through their influence on sediment erosion, and the consequences for intertidal ecology and morphology, can also be
driven by climatic factors such as changes in water and sediment temperature (Marani et al., 2007, 2010; Mariotti and
Fagherazzi, 2012), which is strongly regulated by the light availability due to the turbidity of the water column. Quantifying
and understanding these benthic processes in order to parameterize them into mathematical models is critical for providing
insights into the relative importance of biological and physical factors in sediment erosion/accretion in the intertidal zone.

A range of hydro-morphodynamic models have attempted to parameterize eco-engineering processes on varying spatial and
temporal scales (Brückner et al., 2020; Brückner et al., 2021; Coco et al., 2013; Le Hir et al., 2007; Mariotti and Canestrelli,
2017). While extensive field and flume studies are available in literature on the effect of MPB and faunal on sediment
erosion (Le Hir et al. 2007, Cozzoli et al., 2019), the main challenges in modelling these type of environments is the
complexity of the interaction between the different biotic and abiotic contributors, the time and spatial scales, and the fact
that variation in sediment stability might reflect site-specific differences (Le Hir et al., 2007; Pivato et al., 2019). In fact, the
interactions between these processes are strongly regulated by spatio-temporal conditions, (e.g. Widdows et al., 2000; Van
de Lageweg et al. 2017; Paterson et al. 2018; Best et al. 2018; Cozzoli et al., 2019), making it difficult for predictive models
of sediment stability to make generalities from site-specific findings and to be sufficiently accurate.

For the first time, this study investigates the effect of the environmental conditions, such as temperature, seasonality and
sediment rheology, on biofilm growth and its feedback to the bed stability and morphological evolution over an entire
intertidal channel. The main objective was to investigate and define the key parameters of the biofilm development model
that influence the intertidal channel morphology. The combined effect of temperature, biofilm growth rate and surface
biofilm removal due to tidal dynamic is investigated for different scenarios.





A one-dimensional eco-morphodynamic shallow water model is implemented and tested in this study to investigate the effect of biostabilisation due to the presence of surface biofilm. The model accounts for the effect of tidal oscillation on a non-uniform non-cohesive sediment channel subject to tidal fluctuations at the ocean boundary, and it allows to store the information of the stratigraphy of the deposit emplaced. The biofilm logistic growth model accounts for the effect of hydro-climate variation on the biofilm development, such as temperature changes and carpet-like erosion, as these are key factors controlling biofilm development (Pivato et al., 2019). The model is tested for different benthic biofilm growth rates. Biostabilisation from presence of surface biofilms is implemented in the 1D morphodynamic shallow water model assuming a linear relationship that correlates the amount of biofilm biomass with the increase of the sediment critical shear stress for erosion (Le Hir et al. 2007). The model is applied to an initial flat bed to investigate the implications of different sediment temperatures, representative of different climate scenarios, and different sediment rheology on the channel development.

## 1.1 Bio-sedimentology summary of processes and controls

Since the living and abiotic elements vary temporally and spatially, it is not surprising that the functions and importance of these various factors in determining sediment stability also vary (Black, 1997; Defew et al., 2002; Friend et al., 2003; Paterson et al., 1994; Riethmuller et al., 2000; Underwood et al., 1995; Yallop et al., 1994b). Benthic biofilms change the fundamental properties of sediment and bed substrate: when biofilm develops on the bed surface, it acts as a protective skin on the sediment surface inhibiting entrainment (Paterson et al., 2000); with greater volumes of biofilm required to stabilise sandier beds (Hope et al., 2020).

Numerous studies in marine intertidal environments show a positive correlation between sediment stability in terms of critical shear stress for erosion ($\tau_{bc}$) and EPS components of biofilm. Although EPS is what stabilises the bed, not the MPB themselves, Chlorophyll-a (Chl-a), an indicator of living microphytobenthic biomass, provides a good approximation of biostabilisation potential (Defew et al., 2002; Paterson et al., 2000; Riethmuller et al., 2000, Haro et al., 2022). Chl-a is often the preferred measurement, due to its ecological significance and the fact that it is easy to evaluate (both in the field and by optical remote sensing) (Andersen, 2001; Le Hir et al., 2007). However, the relationships detected between the increase of Chl-a and the increase in bed stability are often weak, emphasising the complexity of this phenomenon and that important interactions are being missed. Hydro-sedimentary processes, modulated by the shear stresses at the bed due to tidal and waves, regulate the biofilm resuspension process and its flux in the water column. The erosion fluxes depend on the bed erodibility, described by the resistance of the sediment to be eroded (Orvain et al., 2014). Changes in bed erodibility, which vary largely in space and time, is the result of a complex interaction between sediment properties, bioturbation activities, grazing, biofilms deposition, reseeding and growth rate (Wood and Widdows, 2002; Thrush et al., 2012; Cozzoli et al., 2019). Due to the complexity of these systems, multiple factors play a relevant role in defining a relationship between critical shear stress for erosion and Chl-a, and there is not a broad relationship but only a general tendency for shear stress to increase with Chl-a content (Paterson et al., 1994; Yallop et al., 1994a; Underwood et al., 1995; Riethmuller et al., 2000; Defew et al., 2002; Friend et al., 2003; Le Hir et al., 2007; Righetti and Lucarelli, 2007; Fang et al., 2014), and often results





are site specific (Riethmüller et al., 2000; Le Hir et al., 2007; Katz et al., 2018). There is thus a fundamental need for a broad-scale bio-morphodynamic approach to synthesis the general effects across habitats modulated, for example, by the distribution of benthic macrofauna, the sediment types, the water content, or the tidal range.

The development of biofilm is controlled by various sedimentary characteristics, biogeochemical drivers, and light-related photosynthesis parameters (e.g. optimum and maximum temperature for MPB photosynthesis, light saturation parameter)
and their spatio-temporal variability (MacIntyre et al., 1996; Pivato et al., 2019, Savelli et al., 2020), the availability of nutrients (Hillebrand and Sommer, 1997), hydrodynamic disturbances such as currents and waves (Mariotti and Fagherazzi, 2012; Tolhurst et al., 2009; Tolhurst et al., 2006), and grazing benthic macrofauna (Hillebrand et al., 2000; Montserrat et al., 2008). However, the prevailing environmental conditions can significantly influence biostabilisation processes. The temperature of the water and underlying sediment layers exerts a major influence on chemical and biological processes and
kinetics as well as benthic nutrient cycling (Smith, 2002, Pivato et al. 2019). In shallow water environments, the energy exchange at the water-sediment interface, the turbidity of the water column and the light reaching the bed surface are crucial to appropriately describe the sediment temperature (Pivato et al., 2018, 2019). Experimental studies of the response of biofilm communities to water warming have shown faster biofilm growth with the increase of temperature (Majdi et al., 2020). Therefore, seasonal temperature changes influence the resistance to erosion (Thom et al., 2015), for example
increases in temperature during the spring, promote photosynthesis, leading to higher Chl-a concentrations and biofilm growth and greater biostabilisation (Underwood and Paterson, 1993; Savelli et al., 2018; Pivato et al., 2019; Haro et al., 2022).

When sediments are covered by biofilm, the entrainment process can occur as sediment-biofilm coated particles (flocs), or via the resuspension of sediment-biofilm aggregates (biomat failure due to carpet-like erosion) (Shang et al., 2014; Fang et
al., 2016; Fang et al., 2017). Resuspended biofilm coated particles can be transported as bedload, and deposited under the different settling velocities, governed by the sediment shape and size, amount of biofilm and density of the particles or flocs (Koh et al., 2007). Hydrodynamic disturbances from currents, tides and waves play a cardinal role, eroding the biofilm and eventually detaching it from the sediment surface. Once the protective biofilm is broken or removed, the underlying clean sediment is exposed, which erodibility is regulated by the characteristic sediment grain size of the substrate (Defew et al.,
2002; Le Hir et al., 2007; Sutherland et al., 1998).

## 2. Methodology

A 1D morphodynamic model for tide-dominated channels implemented with a function that describe the surface biofilm growth was used to determine the relative importance of different bio-physical factors on the development of an intertidal channel longitudinal profile and stratigraphy. The abiotic physical processes included in this study are tidal currents, and
sediment erosion, transport and deposition. The model takes into account the dynamics of biofilm development and its feedback on the erosional and depositional sediment transport processes. The model is based on the one-dimensional shallow





water equations (1D-SWE) for the flow mass, sediment and momentum conservation, modified according with Defina (2000) to account for partially dry areas, such as the beach that can be formed at the landward boundary of the model domain (Figure 1). The model is implemented with a procedure that stores and access the information of the grain size of the

stratigraphy of the deposit.

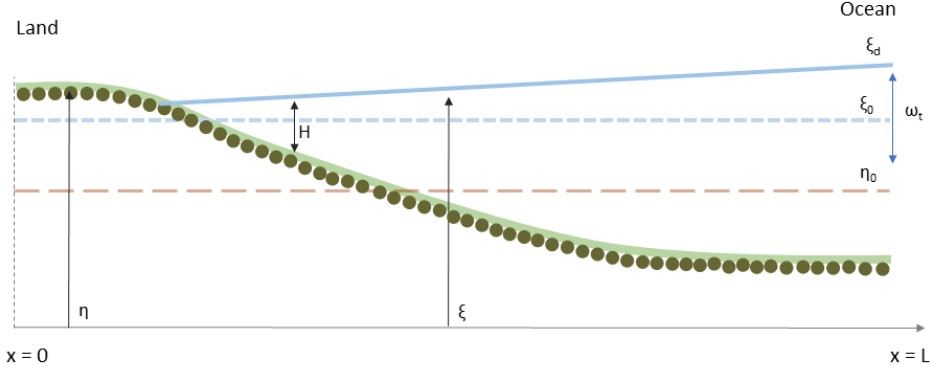

**Figure 1: Schematic representation of the model geometry. The ξ and η are the water surface elevation and channel bed elevation, at the beginning of the simulation the bed is assumed horizontal (η₀) and the mean water surface elevation is ξ₀. ω_t represent the tidal amplitude, and the water surface elevation at the ocean boundary (x = L) is ξ_d**

**2.1 1D SWE model for tidal channel accounting for partially dry areas**

Shallow water equations model (Chaudhry, 2008) are used to describe temporal and 1D spatial variation of an idealised tide dominated channel (Figure 1). The domain is bounded by the ocean, where the tidal oscillations are modelled as a sine curve with amplitude $\alpha_t$ and period $\omega_t$. Input of riverine water and sediment at the landward boundary (Lanzoni and Seminara, 2002), and interaction of the channel with tidal flats and intertidal areas (Todeschini et al., 2008) are assumed negligible.

The shallow water equations, modified by Viparelli et al. (2019) according with Defina (2000), account for the partially dry areas such as when the channel bed is only periodically submerged. Defina (2000) derived the two-dimensional shallow water equations by averaging the Reynolds equations over the bottom irregularities; and then integrated them for mass and momentum conservation in the direction normal to the channel bed. The one dimensional form is obtained by integrating the equations in the transverse direction (Viparelli et al., 2019), giving:

$$
\begin{cases}
F_H \dfrac{\partial A_i}{\partial t} + \dfrac{\partial Q}{\partial x} = 0, & \text{(1a)} \\[2ex]
\dfrac{\partial Q}{\partial t} + \dfrac{\partial}{\partial x}\left(\dfrac{Q^2}{A_c}\right) + \rho g A_c \dfrac{\partial \xi}{\partial x} + \dfrac{\tau_b}{\rho}\chi = 0, & \text{(1b)}
\end{cases}
$$

Where Q is the volumetric flow discharge, $A_C$ is the cross sectional area, and $\rho$ is the water density. The cross sectional area averaged over bed irregularities $A_i$ is equal to (W · ξ), the wet fraction of the channel bed ($F_H$) is computed as function of the characteristic length scale of the bed irregularities ($a_r$, assumed equal to 1 cm), the effective flow depth (Y), and the average bed shear stress ($\tau_b$) acting over the wetted perimeter χ (see Viparelli et al., 2019 for further details of the 1D





morphodynamic model). The model validation is presented in Appendix 1 and shows that the model can reasonably capture

the magnitude and timing of the bed changes. The numerical model is demonstrated to be second order accurate and model

parameters are reported in Table 1.

**Table 1: Parameter for the shallow water model**

| Variable | Value | Description |
|---|---|---|
| L | 25 m | Channel length |
| W | 0.30 m | Channel width |
| $C_f$ | 0.009 | Friction coefficient |
| $D_g$ | 0.3 mm | Geometric mean sediment grain size |
| $\rho_s$ | 2650 Kg/m$^3$ | Density of the sediment |
| $\alpha_t$ | 0.025 m | Tidal amplitude |
| $\omega$ | 12 h | Tidal period |
| $\eta_o$ | 0.4 m | Initial bed elevation |
| $S_f$ | 0 | Initial bed slope |
| $\xi_o$ | 2 m | Mean water surface elevation |
| N | 51 | Number of computational nodes |

### 2.2 Sediment transport model

A sediment transport model is incorporated to describe well mixed, non-cohesive sediment transport and the coupled

morphodynmics (Viparelli et al., 2019). The total volumetric bed material load ($Q_b$) is calculated as the contribution of

bedload and suspended load. The equations to compute the bedload and the suspended load implemented in the model have

been selected to let the direct correlation between the amount of biofilm biomass on the bed, and the updated critical shear

stress for sediment motion that results in biostabilisation.

The bedload is computed using the Ashida and Michiue relation, while the McLean formulation is used to model the

entrainment of sediment in suspension. The total material load ($Q_b$) is the sum of the contribution of bedload and suspended

load, summed over all the grain sizes; and the volume fraction content of sediment with characteristic diameter $D_i$ can be

computed as ($Q_{b,bi}$ + $Q_{b,si}$)/$Q_b$.

The equation for the conservation of the sediment material coupled with a procedure to store the information of the

stratigraphy of the deposit are solved to compute the temporal evolution of the bed profile ($\eta$) and the spatial distribution of

the sediment size (Viparelli et al., 2010). To solve this equation, according with the Hirano active layer approximation

(Hirano, 1971), the deposit can be divided into two regions, the active layer and the substrate. The active layer ($L_a$) is the

topmost part of the deposit where the sediment particles can interact with the flow and it is assumed well mixed, so that the

grain size distribution of the sediment on the active layer can change in space and time but it is assumed constant in the

vertical direction. The substrate ($\eta - L_a$) is located below the active layer and does not interact with the flow; the sediment

fraction in the substrate varies in space, but not in time. Exchange between the substrate and the active layer occurs in the





case of aggradation and degradation. During aggradation the distance between the substrate and the active layer increases, and layers can be added to the grid for the storage of the newly deposited sediment. The grain-size distribution of the antecedent storage layer is computed a weighted average, while the sediment composition of the new storage layers has the same grain-size distribution of the newly deposited material. Interested readers may refer to Viparelli et al. (2010) for further

details about the deposit storage procedure.

### 2.3 Biofilm-dependent erodibility

The novelty of this work is the implementation of a 1D morphodynamic model for intertidal channels with a biofilm growth model that accounts for the effect of seasonality on sediment temperature and light. This study aims to understand the general behaviour of the system and investigate the sensitivity of the biofilm model parameters on the channel development

process, hence the assumption of spatially homogenous biofilm or constrain the development of biofilm only in the cells where the water depth is smaller than 0.05 m are reasonable. Once the biofilm biomass is estimated according with the biofilm growth model, the critical shear stress for erosion is updated to account for the biostabilisation. According with Le Hir et al. (2007) the increase in critical shear stress is assumed proportional to the biofilm biomass available on the bed (B, measured in mg Chl-a/m²):

$$\tau_{bc} = \tau_{bc,0} + \alpha B, \tag{2}$$

Here ($\tau_{bc,0}$) is the critical shear stress for clean sediment. The updated value for the critical shear stress is used in the bedload and suspended load equations to correlate the sediment mobility with the amount of surface biofilm. The time evolution of biofilm biomass (B) is estimated by a simplified model proposed by Mariotti and Fagherazzi (2012) that assumes a logistic grow function for the biofilm biomass:

$$\frac{dB}{dt} = P^B B \frac{1}{1+K_B B} - \varepsilon(B - B_{min}) - E, \tag{3}$$

where $P^B$ is the effective maximum growth rate; $K_B$ is the half-saturation constant which represents the biofilm concentration at which it is reached half of the maximum growth rate and this term accounts for the effect of density limitation. The second term of the equation accounts for the chronic and self-generated biofilm detachment ($\varepsilon$: global decay parameter), not associated with the simulated hydrodynamics (e.g. senescence, heterotrophic processes, benthic macrofauna grazing), and

$B_{min}$ is the amount of background biofilm biomass which allows the recolonization after removal. Starting from a background value for the surface biofilm ($B_{min}$), the biofilm grows only if there are no disturbances limiting the establishment of biofilm. The last term of the equation (3) takes into account the effect of extremely high intensity flow events (E) that are able to mobilize the bed and completely remove the surface biofilm, exposing the clean sediment underneath. At the initial stages, the growth is approximately exponential then, as the saturation begins, it slows to linear

until it reaches maturity when the growth stops and the amount of biofilm on the bed surface remain constant for the entire duration of the simulation, reaching asymptotically an equilibrium condition (Figure 2a).



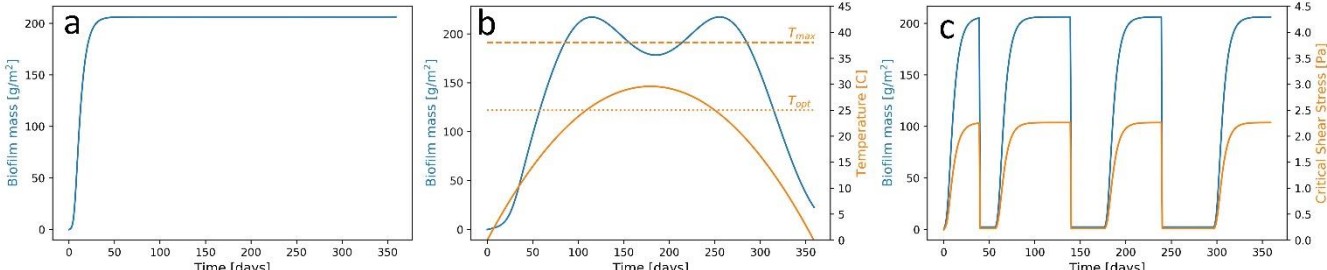

**Figure 2: Biofilm development in time. Biofilm growth pattern in the case of (a) annual undisturbed growing following the logistic grow function, (b) affected by the variation of sediment temperature due to seasonality over a yearlong simulation, and (c) affected by the carpet-like erosion**

The biofilm model has been implemented to account for the seasonal cycle of temperature and light as proposed by Pivato et al. (2019), based on the vertical energy transfer within the water–sediment continuum. This sediment temperature model simulates natural conditions that regulate the development of biofilm, such as the effect of winter conditions that limit the growth of MPB, leading to lower surface sediment biostabilisation and resistance to erosion compared to late spring, summer and early fall, as confirmed also by *in situ* observations (Friend et al. 2002) The MPB photosynthesis and biofilm development are strongly influenced by the seasonal changes of sediment temperature and light availability, which are controlled by the water depth and turbidity (Pratt et al., 2014; Pivato et al., 2019). The sediment temperature model used in this study include these parameters and account for the effect of seasonality, and the maximum growth rate of MPB ($P^B$) is computed according to (Guarini et al., 2000):

$$P^B = P^B_{max} \tanh(H_{res}/E_k), \tag{4}$$

The light saturation parameter $E_k$ (W m$^{-2}$) is assumed constant. The light availability ($H_{res}$) is represented by the residual solar radiation reaching the bed and not reflected by the water surface albedo (A = 0.04) and it is computed as:

$$H_{res} = R_0\, e^{-\lambda Y} \;\; ; \;\; R_0 = (1 - A)\, R_{sun}, \tag{5}$$

The extinction coefficient $\lambda$ represent the capability of the water column to absorb the solar radiance, describing the average effect of the turbidity in the water column (Y: water depth) on radiative transfer, and $R_{sun}$ is the solar radiation. $P^B_{max}$ (h$^{-1}$) represents the growth rate under light saturation conditions, this parameter varies in time and it depends on the surface sediment temperature ($T_{s0}$) according to:

$$\begin{cases} \text{if } T_{S0} < T_{max}: & P^B_{max} = P_{max} \left(\frac{T_{max} - T_{s0}}{T_{max} - T_{opt}}\right)^\beta \exp\left[\beta\left(1 - \frac{T_{max} - T_{s0}}{T_{max} - T_{opt}}\right)\right], \\ \text{if } T_{S0} \geq T_{max}: & P^B_{max} = 0, \end{cases} \tag{6}$$

Function of the optimal and maximum temperature for photosynthesis ($T_{opt}$ = 25 °C, and $T_{max}$ = 38 °C), where the shape factor ($\beta$) is site dependent. The parameter $P_{max}$ represents the maximum value for $P^B_{max}$ and it is site and time dependent. The seasonal changes of the sediment temperature modulates the amount of biofilm biomass, and as a consequence, the





biostabilisation of the bed (Figure 2b). For simplification, in this study, the sediment temperature will be assumed following
a parabolic trend during the one-year interval (blue continuous line in Figure 2b, Pivato et al., (2019)). Biomass increases
exponentially at the beginning of the year, reaching its maximum when the sediment temperature is equal to the optimal

temperature for photosynthesis (dotted orange line in Figure 2b, $T_{opt}$), during spring and fall. As the sediment temperature
increases during the summer months (continuous orange line in Figure 2b), photoinhibition can occur and the biofilm
biomass decreases (blue line in Figure 2b) reaching a local minimum when the sediment temperature is at its maximum and
close to the maximum temperature for photosynthesis (dashed orange line in Figure 2b, $T_{max}$). The growth rate during these
months is still sufficient to enable a fast recovery of the biofilm. As light and sediment temperature decreasing during the

winter seasons, the environmental condition are less favorable for the growth of biofilm. In cases when availability of light at
the bed is limited and the sediment temperature is lower than the optimal temperature for photosynthesis, surface biomass
decreases.

The quantification of the removal of the surface biofilm by intense hydrodynamic forces (carpet-like erosion) occurs in a
very short period of time and so it can be considered as instantaneous, and the catastrophic erosion (E) is:

$$E(B, t) = E_0(B) \sum_i \delta(t - t_i), \tag{7}$$

Where $\delta$ is the Dirac function and $t_i$ is the time of the detachment, $E_0$ is the intensity of the extreme event, assumed as 'all-or-
nothing' process and it can be described as a function of the shear stresses acting on the bed ($\tau$):

$$E_0 = \begin{cases} 0 & \tau \leq \tau_{bc} \\ B - B_{min} & \tau > \tau_{bc} \end{cases}, \tag{8}$$

In the case that shear stresses due to the hydrodynamic forces ($\tau$) are smaller or equal to the value of the sediment critical

shear stress for erosion, there is no disruption of the surface biofilm. In the case that the stress on the bed exceed the critical
value for erosion ($\tau_{bc}$), the biofilm is completely detached and it is reduced to the background value $B_{min}$, exposing the bare
sediment (Figure 2c). When biofilm is removed from the bed surface as carpet-like erosion, the resistance of the bed reduces
to the initial value (Figure 2c) due to the assumption of linear relationship between surface biofilm biomass and critical shear
stress for erosion (Le Hir et al., 2007). This simplified model assumes that in the case of extreme hydrodynamic events, the

erosion is on the order of mm-cm which is much larger than the thickness of the biofilm thickness (μm-mm). Values for the
biofilm model parameters are reported in Table 2.

**Table 2: Biofilm model parameter**

| Parameter | Description | Value |
|-----------|-------------|-------|
| $\varepsilon$ | Global decay | 0.2 day$^{-1}$ |
| $P_{max}$ | Maximum growth rate | 1.07 day$^{-1}$ |
| $K_B$ | Half-saturation constant for biofilm growth | 0.02 (mg Chl-a/m$^2$)$^{-1}$ |
| $B_{min}$ | Background biofilm | 1 mg Chl-a/m$^2$ |
| $E_k$ | Light saturation parameter | 100 W m$^{-2}$ |





| $T_{max}$ | Maximum temperature for photosynthesis | 38 °C |
| $T_{opt}$ | Optimal temperature for photosynthesis | 25 °C |
| $\beta$ | Shape parameter | 2 |
| A | Water surface albedo | 0.04 |
| $R_{sun}$ | Solar irradiance reaching the water surface | 6.33 $10^7$ Wm$^{-2}$ |
| $\lambda$ | Extinction coefficient | 2.0 m$^{-1}$ |
| $\alpha$ | Bio-cohesivity paramenter | 0.01 Pa/(mg Chl-a/m$^2$) |
| $\tau_{bc,0}$ | Clean sediment critical shear stress (without biofilm) | 0.2 Pa |

The sediment mixture used for the simulations is characterized by median diameter $D_{50}$ = 0.323 mm and geometric mean sediment grain size Dg = 0.303 mm.

This study investigates the sensitivity of the key biofilm model parameters. Firstly, it is presented a sensitivity analysis of the stability of the biofilm by changing the biofilm model parameters and by simulating different hydrodynamic disturbances characterized by periodicity (T) and shear stress ($\tau_0$). Secondly, different scenarios for bio-modulation of channel morphodynamics are explored: i) carpet-like erosion; ii) seasonality; iii) biofilm growth rate; iv) sediment temperature variation; and v) bio-cohesivity of the sediment ($\alpha$). The numerical simulations have been performed in the absence of an

imposed input of sand from the ocean, and without riverine water and sediment at the landward boundary. It has been assumed semidiurnal tidal using an idealized 25 m long channel with constant width equal to 30 cm.

## 3 Results

### 3.1 The control of hydrodynamic disturbances and biofilm model parameters on biofilm stability

The combine effect of seasonality in sediment temperature and hydrodynamic events are reported in Figure 3, under a set of

different temperature-influenced scenarios that are intended to simulate the changes in nutrient availability in the water (growth rate parameter) and the long term variation of temperature. The reference profile for the development of biofilm is reported in Figure 2b ($T_{s0,max}$ = 32 °C, $P_{max}$ = 1.068 days$^{-1}$).



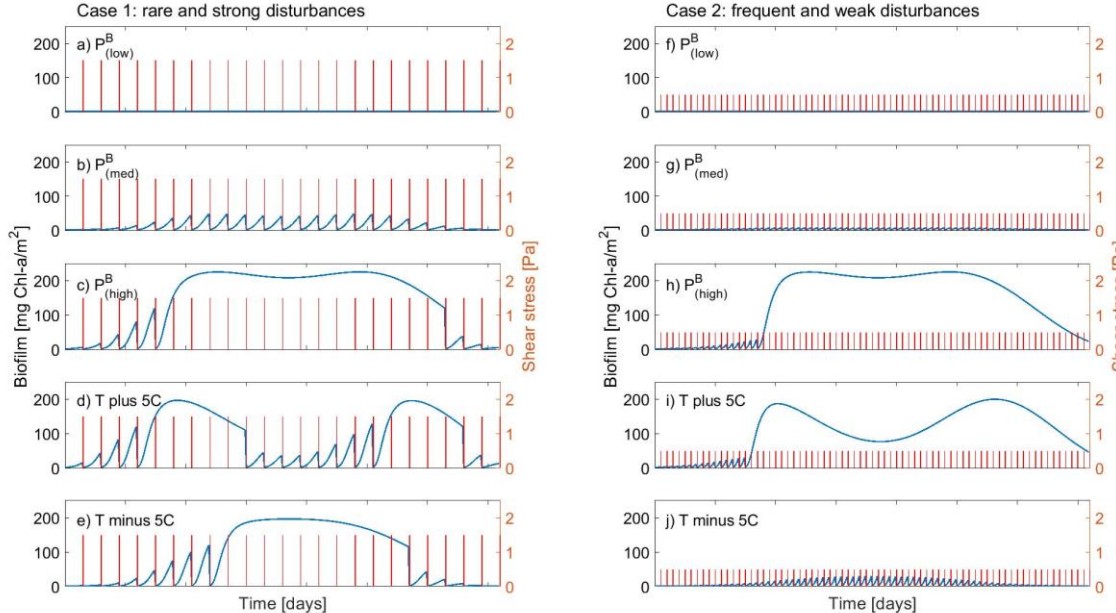

**Figure 3: Effect of seasonality and hydrodynamic forces on the evolution of surface biofilm biomass. The evolution of the**
**temperature of the sediment at the bed is simulated for a period of one year, under rare (every 15 days) and strong (1.5 Pa)**
**hydrodynamic disturbances. The effect of different values of the growth rate parameter (panels a, b c) and sediment temperature**
**are investigated (panels d and e). The evolution of the temperature of the sediment at the bed is simulated for a period of one year,**
**under frequent (every 5 days) and weak (0.5 Pa) hydrodynamic disturbances. The effect of different values of the growth rate**
**parameter (panels f, g, h) and sediment temperature are investigated (panels i and j).**

With high-intensity and rare events (case 1, T = 15 days, $\tau_0$ = 1.5 Pa), and small values of the grow rate parameter ($P^B_{max}$ =
0.0078 and 0.5617 days$^{-1}$), the new settled biofilm is periodically detached by the disturbances (Figure 3a and Figure 3b).
Biofilm grows during the time span between two consecutive events (Figure 3a-b), but it is destroyed every time a
significant hydrodynamic event occurs. The increase in sediment resistance is not enough to prevent the erosion caused by
the high-intensity events. A further increase of the maximum growth parameter ($P^B_{max}$ = 1.068 days$^{-1}$) results in a more rapid
growth and establishment of biofilm. During the initial and final months of the simulated year (January to mid-March, and,
after mid-November) the biofilm is periodically removed, because the temperature of the sediment inhibit the development
of biomass (Fig. 3c). On the other hand, during spring and summer months the combination of the temperature conditions
and the high growth rate promote the development of stable biofilm which is able to resist the periodic disturbances.

The biofilm biomass profile under rare and intense hydrodynamic disturbances with a variation of the annual sediment
temperature by ±5 °C compared with the previous simulation ($P^B_{max}$ = 1.068 days$^{-1}$), is reported in Figure 3d-e. In the case of
an increase of the sediment temperature the profile, analogously to the previous cases, show a slowdown of the development
of biofilm in winter and fall. Furthermore, during the summer period (June to September), the sediment temperature
increases above the optimal temperature for photosynthesis ($T_{opt}$ = 25 °C) resulting in a drop in EPS production, reducing the
bed stabilization (linear decrease of the critical shear stress for erosion) and becoming more vulnerable to the hydrodynamic



disturbances (Fig. 3d). The effect of an overall annual reduction of the sediment temperature on biofilm is shown in Figure
3e. In this case the biofilm is more vulnerable to the disturbances at the beginning and at the end of the simulated year
compared with the profile in panel c, due to the fact that temperature conditions further from the optimal temperature for
photosynthesis reduces the rate of development of biofilm.

When biofilm growth rate parameter is low (Figure 3f and Fig. 3g), under frequent and weak disturbances (case 2, T = 5

days, $\tau_0$ = 0.5 Pa), biofilm is periodically detached and it cannot establish during the entire simulated year. An increase of the
growth rate parameter shows that in summer the biofilm can establish and cover the bed surface until the end of the year,
even though the biomass decreases in fall and winter (($P^B_{max}$ = 1.068 days$^{-1}$, Figure 3h), unlike the case of strong
disturbances (panel c). The increased amount of biofilm enhanced the bed stabilization inhibiting the erosional behaviour
also under further disturbances. Comparing Figure 3h with the case in which the annual sediment temperature is increased

(Figure 3i) or decreased (Figure 3j) by 5 °C show that an increase in temperature would decrease the amount of biomass at
the bed. On the other hand, a decrease of sediment temperature would not allow biofilm to establish because it would be
constantly destroyed by the frequent disturbances.

It is reasonable to conclude that the presence of consolidated biofilm able to stabilize the bed does not only depend on the
intensity and the frequency of the disturbing events, but also sediment temperature and seasonal parameters play a key role.

The amount of biofilm biomass on the bed surface plays a significant role in defining areas of erosion, even under the same
hydrodynamic conditions (Mariotti and Fagherazzi, 2012; Hope et al. 2020).

The sensitivity analysis of the biofilm model parameters (Equation 2-3-4-5-6) have been investigated by systematically
changing the intensity and the periodicity of the disturbances, to find the hydrodynamic conditions at which the status of the
biofilm change from stable to detached (Fig. A3 appendix). Mariotti and Fagherazzi (2012) have shown that biological

biofilm growth parameters ($P^B_{max}$, $K_b$, $\varepsilon$) can affect the stability of surface biofilm in terms of the resistance of biofilm to be
eroded from the bed by high intensity hydrodynamic forces, i.e. tides, as well as the proportional parameter that describes the
effect of biofilm on sediment strength ($\alpha$). The parameters that describe the effect of the light availability as the water surface
albedo (A), the dimensionless shape factor in the equation that describes the sediment temperature ($\beta$), the extinction
coefficient ($\lambda$) which proxy of the water column turbidity and the light saturation parameter ($E_k$) do not influence the growth

of biofilm under the effect of different hydrodynamic disturbances. The proportional coefficient that correlates the presence
of surface biofilm with the increase of the bed resistance results to be important in the determination of the equilibrium
configuration (steady biofilm).

**3.2 Effect of seasonality and carpet-like erosion**

This section explores the effects of seasonality and carpet like erosion on the morphological evolution of an intertidal

channel. Two main cases are considered in this study for the spatial distribution of biofilm. One case assumes that biofilm is
uniformly distribute in the entire computational domain (left two panels in Figure 4), to explore the case of biofilm
development also in the deepest portion of the channel as it has been also suggested in literature. In fact, biostabilising





organisms are found along the entire tidal range, from intertidal and subtidal areas, to shellfish reefs and on the continental shelf (Cahoon, 1999; Pinckney, 2018; van de Vijsel et al. 2020). For the second set of simulations, the biofilm is assumed to

grow in turbid systems, where light attenuation would prevent substantial growth of surface biofilm due to the limited availability of light for the photosynthesis processes. The development of biofilm is therefore limited on locations where the water depth is below 0.05 m, which corresponds to the portion of the channel that experience the wet-dry transition according with the tidal amplitude range used for these simulations (two right columns in Figure 4).

The model is applied to investigate the separate and combine effect of carpet-like erosion due to hydrodynamic forces and

seasonality. The bed evolution (first and third column) and the stratigraphy of the deposit (second and fourth column) after 30,000 tidal cycles are reported in Figure 4, where for the bed evolution profiles the green dashed line represent the initial bed, and the blue dashed line the initial mean water surface. The profiles are compared to the final equilibrium bed profile in the case of clean sediment (Figure 4a, red dashed line). In the second and fourth column of Figure 4 are reported the spatial distribution of the geometric mean diameter of the deposit at the end of the simulation. Initial mean diameter of the

transported sediment and of the bed is 0.3 mm.

The model initial conditions assume a flat bed, there is a formation of an upstream migrating shore at the landward boundary due to the effect of tides at the ocean boundary, creating an alluvial deposit characterized by sediment erosion at the ocean boundary (Lanzoni and Seminara, 2002; Tambroni et al., 2005; Todeschini et al., 2008; Viparelli et al., 2019). As the shoal reaches and is impeded at the landward boundary, a beach forms and grows until conditions of morphodynamic equilibrium

are met (approximately after 20,000 tidal cycles). Grain size distribution of the landward deposit associated with shoal reflection coarsened in the upward direction and from the ocean to the land (Figure 4b). Coarse sediment is transported upstream of the shoal, and is deposited in the landward part of the channel forming the coarse basal part of the deposit. As the shoal approached the landward boundary, fine sediment is deposited on the basal layer. Sediment deposited after the shoal reflection presented a fining upward profile for decreasing velocities associated with beach formation.

First, the case of spatially-uniform and stable biofilm on the bed surface for the entire duration of the simulation (insert of Figure 4c) is modelled. This resulted in less sediment mobility compared to the clear sediment scenario (Figure 4a). The bed exhibits minor erosional behaviour at the ocean boundary, while at the land boundary the bed profile does not change in time and the bed is stable. The sediment mean diameter at the bed surface at the ocean boundary is coarser than the initial condition (Figure 4d). Figure 4e and Figure 4f show the bed elevation and mean diameter of the deposit with the water depth

as constrain for the development of biofilm (H < 0.05 m). The bed is more mobile both at the ocean and landward (Figure 4e), and coarse sediments are deposited upstream (Figure 4f), with the mean diameter at the surface becoming coarser than the initial condition (0.35 mm).



**Figure 4: Bed evolution (column 1) and spatial changes of the geometric mean diameter of the deposit (column 2) after 30,000 tidal**
**390** **cycles, assuming that the biofilm is spatially uniformly distributed on the channel. Column 3 and 4 represent the bed profile and**
**the grain size distribution of the deposit, with biofilm developing only in locations where the water depth is smaller than 0.05 m.**
**The rows represent: (1) clean sediment (benchmark case), (2) stable biofilm, (3) biofilm development regulated by seasonality, (4)**
**biofilm development regulated by carpet-like erosion due to hydrodynamic forces, (5) surface biofilm affected from both**



**seasonality and carpet-like erosion. The bed profiles are compared with the clean sediment final bed elevation at equilibrium (red dashed lines). The blue and green dashed lines represent the initial water surface elevation and the initial bed profile respectively. Initial geometric mean size of the sediment at the bed is 0.30 mm.**

Considering the effect of seasonality (insert in Figure 4g) there are slight increases in sediment mobility seaward (Figure 4g), creating a coarser bed deposit (Figure 4h) compared with the previous case. While, when assuming that biofilm is present only in shallow water conditions (H < 0.05 m), it results in an increase of bed mobility (Figure 4i). Additionally, the bed

needs longer time to reach equilibrium state, so it is reasonable to conclude that, even after 30,000 tidal cycles, the bed profile is still evolving. Coarse sediment is found on the seabed at the ocean boundary, while landward the bed is characterized by fine sediment while the deposit at the beginning of the simulation is preferentially coarse (characteristic mean diameter of 0.35 mm) (Figure 4j).

Figure 4k shows the evolution of the bed with surface biofilm periodically removed (carpet-like erosion) by the tidal induced

stresses on the bed. Hydrodynamic forces play a relevant role in shaping the bed, and the final profile is similar to the benchmark case, with erosion at the ocean boundary and deposition at the land boundary (Figure 4a). The presence of biofilm hinders bed evolution and more time is required to reach the equilibrium state. This is due to the periodic removal of surface biofilm due to the tidal forces, which causes periodical decreases in the bed critical shear stress for erosion and therefore biostabilisation. The stratigraphy of the deposit (Figure 4l) is analogous to the clear sediment case (Figure 4b), with

initial coarse sediment deposited landward until the shore reflects creating a lens of fine material, after that more coarse sediment is deposited. Assuming that biofilm is developed only under shallow water conditions (water depth smaller than 0.05 m), the channel needs more time to reach equilibrium (Figure 4m). This observation is confirmed in Figure 4n as the shore, after 30,000 tidal cycles, did not reached the upstream boundary.

For both the biofilm spatial distribution conditions investigated in this study (uniform biofilm and biofilm only in shallow

water conditions), the effect of both seasonality and carpet-like erosion results in similar bed profile. In fact, the bed morphology is comparable both in terms of bed elevation (Figure 4o and Figure 4q) and stratigraphy of the deposit (Figure 4p and Figure 4r).

### 3.3 Effect of maximum biofilm growth rate parameter

Changes in the maximum growth rate of the biofilm development model ($P_{max}$, equation 6) results in a faster development of

the biofilm, furthermore the peak of biofilm biomass appears in early stage of biofilm development (Figure 5a). In the simulations showed above (Fig. 4), the maximum growth rate parameter has been assumed equal to 1.07 day$^{-1}$, which is a reference value that would give a biofilm biomass of 200 mg Chl-a/m$^2$ in steady state conditions. Figure 5 shows the morphology and the stratigraphy of the final bed, after 30,000 tidal cycles under different values of the biofilm grow rate parameter: small (Figure 5b and Figure 5e, $P_{max}$ = 0.0078 day$^{-1}$), medium (Figure 5c and Figure 5f, $P_{max}$ = 0.56 day$^{-1}$) and a

large (Figure 5d and Figure 5g, $P_{max}$ = 1.10 day$^{-1}$). Surface biofilm in these simulations has been assumed developing only in locations where the water depth is smaller than 0.05 m.





Small or medium values for the maximum growth parameter for biofilm create a similar final longitudinal bed profile, while for large values of $P_{max}$ the morphology of the bed is significantly influenced by the presence of surface biofilm. In the case of small (Figure 5b) and medium (Figure 5c) values of $P_{max}$ the final bed profiles are similar, even if smaller grow rate

parameter results in a slightly higher bed mobility and the bed reaches sooner the final bed equilibrium condition. A large maximum growth rate parameter influence the morphological evolution of the channel by promoting the development of surface biofilm from the early stages of the simulation and reducing the sediment mobility (Figure 5a, green line). Under this condition, the bed after 30,000 tidal cycles still dynamic both at the landward and seaward boundary. Overall, the grain size distribution of the channel bed is preferentially coarse in the seaward boundary, and fine at the landward boundary.

Simulations with small values of the growth parameter result in higher sediment mobility and the deposit at the landward side is relatively coarse (Figure 5e) compared with the stratigraphy of the deposit created in the case of large growth rate parameter. In this case, due to the high stabilization, coarse fraction characterize the bed surface at the sea boundary (Figure 5g).

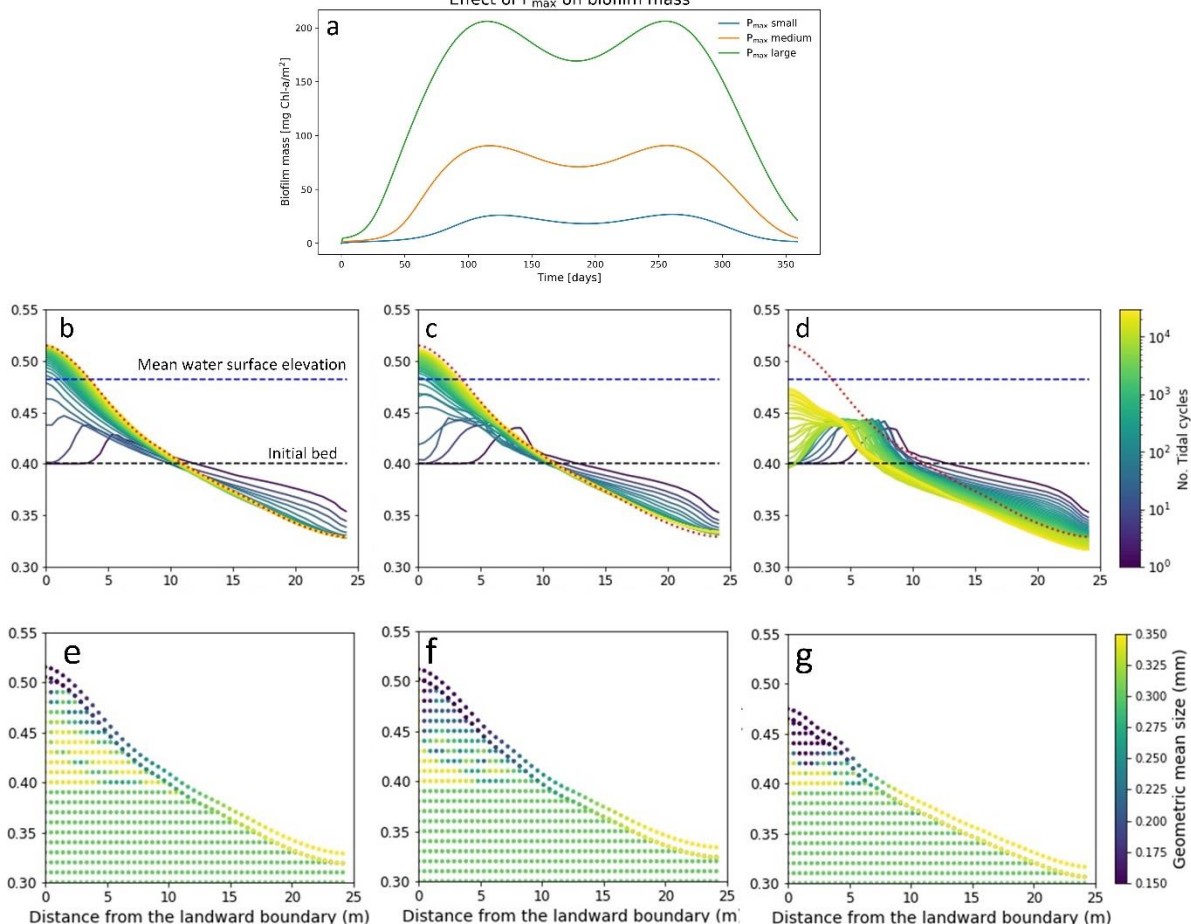



**Figure 5: Effect of different values of maximum growth rate ($P_{max}$) on the surface biofilm biomass (a). Top row represent the bed evolution profile with small (b), medium (c) and large (d) $P_{max}$, after 30,000 simulated tidal cycles. The bottom row represent the geometric mean diameter of the final deposit, in the case of small (e), medium (f) and large (g) $P_{max}$. The bed profiles are compared with the clean sediment final bed elevation at equilibrium (red dashed lines). The blue and green dashed lines represent the initial water surface elevation and the initial bed profile respectively. Initial geometric mean size of the bed is 0.30 mm.**

### 3.4 Effect of temperature variation

Biofilm growth differ during the course of a year due to environmental conditions, with higher growth rate during spring and beginning of summer (Thom et al., 2015; Widdows et al., 2000). The seasonality and the variation of the sediment temperature effect the development of biofilm and the consequent morphological evolution of the channel, as shown in Figure 6a. In the simulations presented here, the high hydrodynamic forces that could completely remove surface biofilm are

neglect (carpet-like erosion).

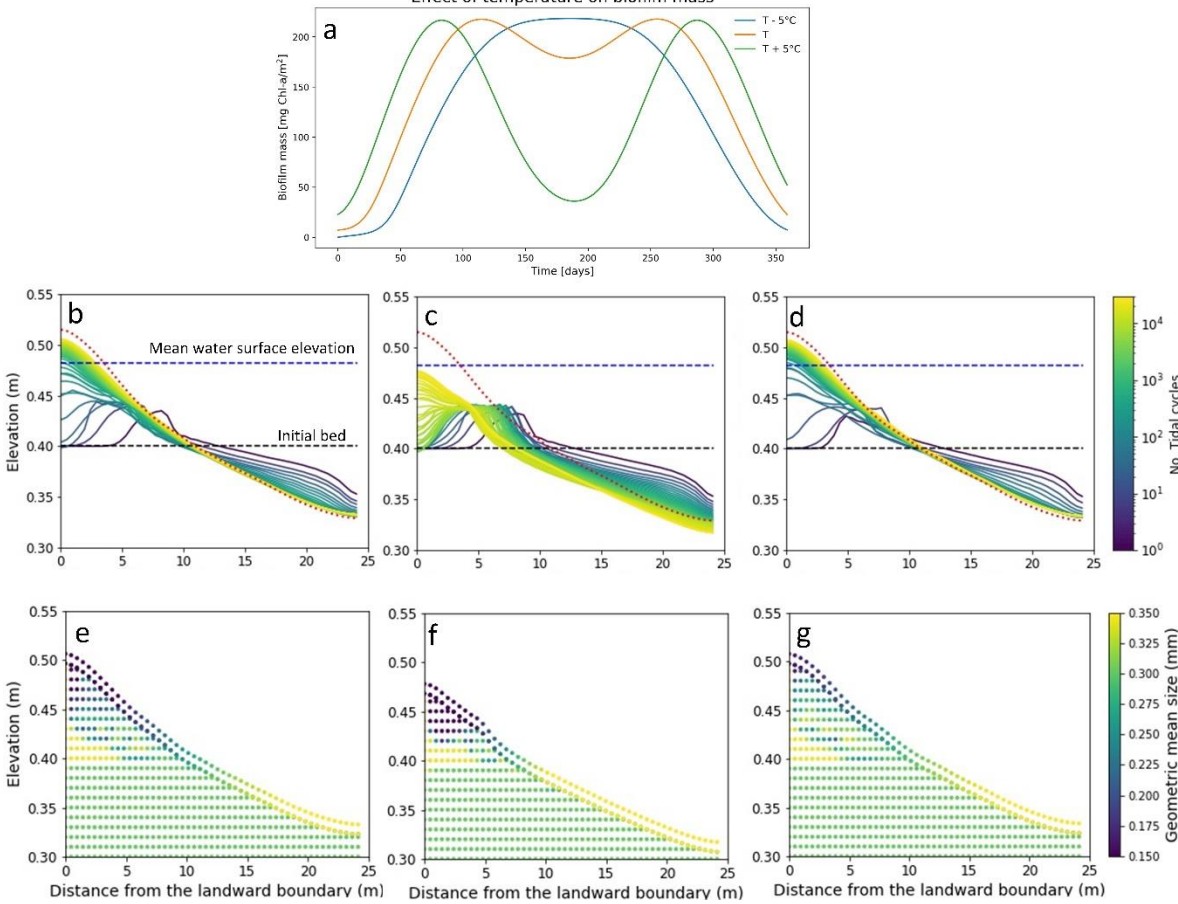

**Figure 6: Effect of sediment temperature on the development of biofilm (a). Bed evolution profile (b, c, d) and final stratigraphy of the deposit (e, f, g) after 30,000 tidal cycles in the case of low sediment temperature (−5°C, left panels), reference case (center panels) and high sediment temperature (+5°C, right panels) respectively. The bed profiles are compared with the clean sediment**
**final bed elevation at equilibrium (red dashed lines). The blue and green dashed lines represent the initial water surface elevation and the initial bed profile respectively. Initial geometric mean size of the bed is 0.30 mm.**





The variation of the sediment temperature are function of the light availability and the turbidity of the water column. Here it is assumed a sediment temperature variation of ±5°C compared to the previously simulated temperature profile, to simulate the possible scenarios in shallow water environments (Pivato et al., 2019). As mentioned before, the amount of biofilm

biomass developed on the bed surface is strongly regulated by the sediment temperature (Figure 6a). Compared with the reference case (orange line), a decrease of the annual sediment temperature (T – 5°C) result in an overall slower development of biofilm, in other words it takes longer for the biofilm to reach the maximum amount of biofilm biomass at the bed (approximately 150 simulated days, blue line Figure 6a). In this scenario the sediment temperature does not reach the maximum temperature for photosynthesis ($T_{max}$) resulting in a stable biofilm biomass over a relative long period (~ between

150 and 250 days). On the other hand, an increase of the annual sediment temperature (T + 5°C) would result in a more rapid development of biofilm compared with the reference case, reaching the maximum amount of surface biomass after approximately 80 simulated days (green line in Figure 6a). The sediment temperature reaches and surpasses the maximum temperature for photosynthesis ($T_{max}$) resulting in a decrease of surface biofilm.

In the reference scenario, the total amount of biofilm biomass covering the bed over the year is comparable to the case of

low sediment temperature (Figure 6a), but these two scenarios result in a slightly different final bed profile. In the case of low sediment temperature the bed is covered by biofilm for the period between 120 and 240 days (May – August), while in the reference case the bed shows presence of biofilm for a longer period of time, even if it is not always at its maximum value (between 90 and 280 days). This result in a more stable bed, morphological changes occur in a longer timeframe and, by the end of the simulation, it does not reach an equilibrium condition (Figure 6c). In the case of low temperature, the bed

reaches a stable profile (Figure 6b) and the stratigraphy of the emplaced deposit is characterized by coarser sediment (Figure 6e) compared to the reference case (Figure 6f). The last simulated scenario characterized by high sediment temperature, show a final bed profile in equilibrium (Figure 6d), in analogy with Figure 6b, confirming that the bed in this case is more mobile, due to the smaller time that the bed was covered by biofilm.

## 3.5 Effect of the sediment bio-cohesivity parameter (α)

There is no universal relationship available in literature between critical shear stress for erosion ($\tau_{bc}$) and the amount of Chl-a, considered as approximation of biostabilisation potential. This uncertainty can be explained by sediment rheology as well as different sampling techniques. Furthermore, the distribution of Chl-a content can vary spatially due to the small scale morphology of the bed (Le Hir et al., 2007).

The effect of this variability has been investigated by changing the parameter (α) used to correlate the critical shear stress for

erosion ($\tau_{bc}$) with the amount of Chl-a on the bed (Eq. 2). The results of the channel morphology and stratigraphy obtained by assuming (α = 0.01) as suggested in literature (Le Hir et al., 2007; Mariotti and Fagherazzi, 2012) are compared with scenario that account for the variability of the sediment bio-cohesivity (Figure 7). The values of the bio-cohesivity parameters tested here (α = 0.005 and 0.015) have been suggested by previous studies and by data from the intertidal flat in the German Wadden Sea by Le Hir et al. (2007).





Small value of the bio-cohesivity parameter (α = 0.005) results in higher channel mobility, that is able to reach equilibrium
by the end of the simulation (Figure 7a). The final bed surface grain size distribution (Figure 7d) is mostly characterized by
fine sediment (D < 0.25 mm), with the coarse fraction covering the landward boundary of the domain (x < 10 m), compared
with the reference case where a larger surface area is covered by coarse material (Figure 7e). A further increase of the bio-
cohesivity parameter results in a slower morphological evolution of the channel. After 30,000 simulated tidal cycles the

channel still evolving in time (Figure 7c) and it results in a coarser bed surface (Figure 7f), as result of a stronger surface
biofilm that inhibit sediment motion.

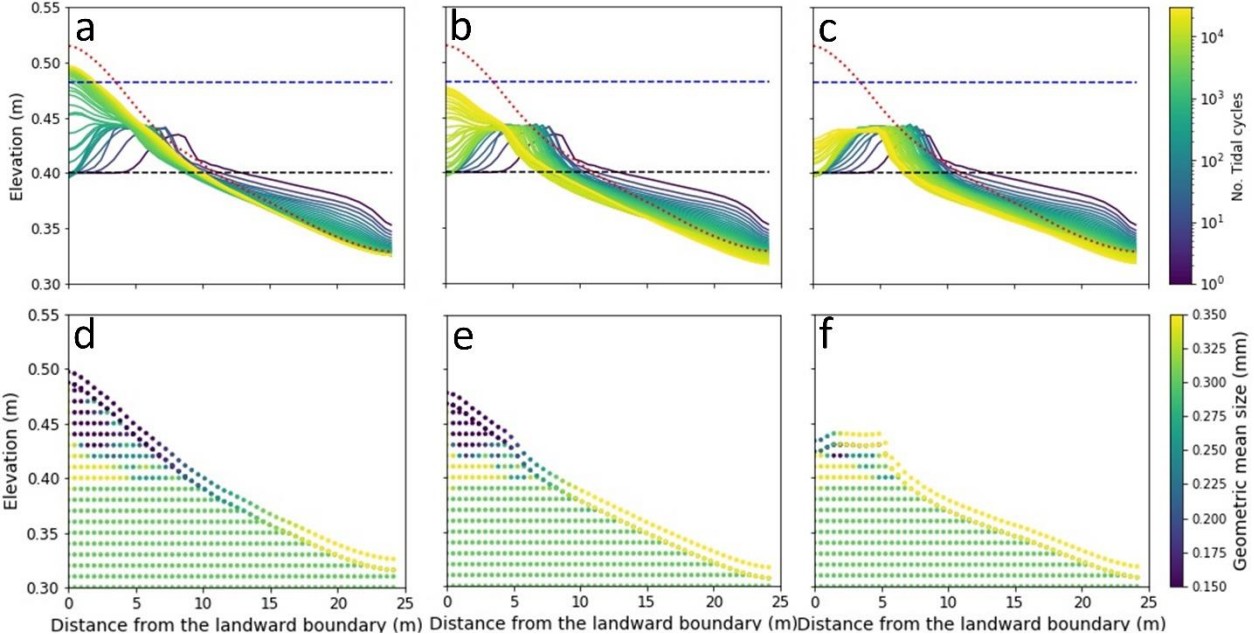

**Figure 7: Effect of bio-cohesivity parameter (α) that relates the biofilm with critical shear stress for erosion. Bed evolution profile
(a, b, c) and final stratigraphy of the deposit after 30,000 tidal cycles (d, e, f) in the case of α=0.005, reference case (α=0.010) and**

**α=0.015 respectively. The bed profiles are compared with the clean sediment final bed elevation at equilibrium (red dashed lines).
The blue and green dashed lines represent the initial water surface elevation and the initial bed profile respectively. Initial
geometric mean size of the bed is 0.30 mm.**

## 4 Discussion

The complex interaction between physical, chemical and biological processes and properties that govern sediment transport

mechanisms are still poorly understood and quantified. It is therefore difficult for morphodynamic models to be accurate and
predict into the future. Whilst some factors will be similar between estuaries, our findings confirm the need for site-specific
calibration of morphodynamic models. These models must account for the contribution of different eco-engineers on tidal
flat development. Nonetheless, our investigation offers both fundamental qualitative and quantitative information regarding
the role of key environmental parameters in sediment stability and morphological evolution in a simplified intertidal channel.



Local hydrodynamic conditions (e.g. tides, waves) not only affect the establishment of biofilms but their recovery processes (Defew et al., 2002). Small, but frequent disturbances hinder the early stages of biofilm development, while strong disturbances can detach established biofilm (Figure 3). It is reasonable to conclude that local hydrodynamics play a crucial role in mediating the presence of biofilm, with carpet-like erosion possible when disturbance is high. Results presented in this study show that in low dynamic environments where carpet-like erosion is not dominant (e.g. on bars, in central areas of tidal flats), biofilms growth is prominent (inserts of Figure 4c and d), resulting in a strong bio-stabilizing effect on the bed (Figure 4c and d). This is in agreement with field investigations where higher bed stability is observed in central tidal flats compared to the edges (Widdows at al., 2000). Biofilm presence inhibited the sediment movement, for all shallow water habitats with low tidal forces, as demonstrated by a lack of significant changes after 30,000 tidal cycles (Figure 4c-i). Furthermore, deposited sediment was coarse. In high dynamic environments, carpet-like erosion can remove surface biofilm exposing the clean sediment underneath and reducing biostabilisation (e.g. close to the channel, at the edge of the tidal flat), resulting in a more mobile bed profile (Figure 4k-q). Moreover, high bed shear stresses due to hydrodynamic forces (tides) can cause a general delay in biofilm formation and biostabilisation (Figure 3) and a significant decrease of the biofilm stability (Schmidt et al. 2018). Simulations presented here demonstrate that the biostabilising effect due to the presence of biofilm decreases the time needed for the bed to reach equilibrium compared to clean, abiotic sediment (Figure 4a). The deposits are finer than the initial bed condition at the landward boundary, which is particularly relevant as physically cohesive sediment, like mud, facilitates saltmarsh survival and MPB growth. This growth in turn promote further sedimentation and can limit mud erosion (Brückner et al., 2020) which is fundamental for stabilization of large estuary morphology, bank accretion and stability, predicting estuarine and deltaic development, and coastal protection. Consequently, an increasing extent and thickness of mud cover might lead to a stabilization of large-scale estuarine morphology. Even when trends are observed between the amount of benthic biofilm and the grain size distribution at the bed, the relationship between these two parameters is not straightforward. These relationships are strongly modulated by the role played by a complex interaction of other factors, such as the light reaching the bottom, the nutrient fluxes and human activities, and community composition of the primary producers present such as diatoms, cyanobacteria and green algae (Cahoon et al., 1999; Schmidt et al., 2018). Furthermore, in energetic and sandy sites, the frequent reworking of the substrate results in removal of the biofilm and more mobile bed; on the other hand, even small increases of fine and muddy sediment fraction can promote sediment stability (Hope et al. 2020).

In aquatic environments, spatial variability in water temperature can be natural (e.g. geothermal activity, source of water) or it can result from direct changes in local land use and activities (e.g. deforestation, industrial activities), or indirect and global changes (e.g. climate change) (Caissie, 2006; Van Vliet et al., 2011). Alteration of thermal regimes can be a major determinant of changes in the diversity and resilience of aquatic biota from primary producers to consumers. The temperature model implemented in this study used surface sediment temperature as a key parameter for the growth of biofilm (Pivato et al., 2019; Pivato et al., 2018). The model is validated for temperate areas (red line in Figure 6a), and tested for ±5°C variation of temperature from the reference case. Favourable temperature conditions would result in changes in





biofilm biomass production and affect fine sediment dynamics by local stabilization and accretion, reduce the turbidity in the

water column, and change the hydrodynamic conditions (reduce the bed roughness). In summary, the goal of these simulations was to investigate the bed morphology in colder and warmer climates, as it is reported in literature. In the first case the activity of MPB is restricted to the warmer periods (blue line in Figure 6a), while in the second case biological biomass at the bed can develop more during cooler months (green line in Figure 6a, Hope et al., 2019). Temperature regulated biofilm development at the bed strongly influence the final morphology of the channel. Dissolved oxygen levels

are directly linked to water temperature, with low values of saturated dissolved oxygen for higher water temperature (Pivato et al., 2019). Projected future temperature increases could thus lead to a decrease in ocean oxygen solubility and have a direct effect on organismal physiology, and a consequent increase of biofilm development especially in shallow water basins located in temperate regions (Kent et al., 2018). Nutrient cycling and carbon flows through benthic communities are influenced by chemical and biological processes, which are regulated by sediment temperature and light availability.

Therefore, here by considering the effect of biostabilisation, this model indirectly account for the effect of water and sediment temperature on the morphodynamic evolution of coastal shallow bays (Marani et al., 2007, 2010; Mariotti and Fagherazzi, 2012).

Biofilm growth rate and seasonality are key parameters when modelling biostabilisation (Figure 4 and Figure 5). Large variation in biostabilisation between seasons is reported in literature with the highest values in spring and the lowest in late

autumn (Underwood and Paterson, 1993; Marcarelli et al., 2008; Thom et al., 2015; Schmidt et al., 2016; Waqas et al., 2020). This is due to the differences in biofilm growth and composition resulting in mechanically diverse responses to the increased bed shear stress. Experiments conducted by Thom et al. (2015) reported a tenfold increase in sediment stability, depending on boundary conditions and investigated season, and the hydrodynamic erosional process can be influenced as well by seasonality, highlighting the heterogeneity of the process. Biostabilisation is considerably higher in spring than in

summer, supported by the fact that EPS protein and carbohydrate contents increase (Amos et al., 2003; Dickhudt et al., 2009; Thom et al., 2015; Schmidt et al. 2016, 2018). Seasonality also affects bed morphology, during early spring until the onset of summer, with 80% of the surface of the intertidal flats covered in biofilm, which can enhance the formation of a hummock-hollow pattern (Weerman et al., 2011a). This trend is observed in temperate humid climate (cold winters and mild summers, Figure 6a blue line, Figure 6b and e), while in warm temperate climate (mild winters and hot summers, Figure 6a green line,

Figure 6d and g) the seasonal MPB biomass maximum is most likely to occur in late fall (Haro et al., 2022). Friend et al. (2003) also observed a strong seasonally dependent relationships between critical shear stress for erosion ($\tau_c$), habitat type, Chl-a, and bed elevation, in fact the seasonal activity of the species contributes significantly in increasing or decreasing the sediment stability (Thom et al., 2015). This aspect has been parametrized in this study in the maximum growth rate parameter ($P_{max}$) which accounts for the effect of seasonality according to a sediment temperature model (data available in

literature assume this parameter ranging between 0.0078 – 1.10 day$^{-1}$) (Labiod et al., 2007; Mariotti and Fagherazzi, 2012; Uehlinger et al., 1996).





Experimental and field studies have attempted to identify the roles of biological and physical processes in sediment stability using regression analyses to relate the erosion threshold to biological and physical parameters (Defew et al., 2003; Amos et al., 2004; Droppo et al., 2007; Grabowski et al., 2011). Chl-a has been found as a good proxy for the sediment critical shear stresses, as a strong functional relationship exists between Chl-a and the amount of biofilm derived from EPS in intertidal surface sediments (Friend et al., 2003; Paterson et al., 1994; Underwood et al., 1995). Currently, there is no universal relationship available in literature for the sediment bio-cohesivity parameter (α) that correlates the critical shear stress for erosion and Chl-a, besides the observation that the critical shear stress for erosion increases as the Chl-a content increases (Le Hir et al., 2007). The presence of biofilm can result in a bed stabilization up to 500% compared with non-colonized sediment le (Le Hir et al., 2007; Zhu et al., 2019), suggesting that the effect of EPS is more important than physical cohesion (Malarkey et al., 2015; Parsons et al., 2016). While it is clear from literature that there is a link between Chl-a and biostabilisation ($\tau_{bc}$), which affect the sediment resuspension and transport in the water column (Le Hir et al., 2007), there still some uncertainties on the factor (α) that correlates these two parameters. This uncertainty can be explained by the fact that the sample technique plays an important role in shaping this relationship, both due to the variety of experimental devises and the measuring method as well as the fact the Chl-a content varies strongly in the bed and geographically. It is acknowledged in literature that there are some uncertainties due to the sampling techniques (Paterson et al., 2000; Widdows et al., 2007) as these studies are conducted in situ, where many uncontrolled parameters are likely to vary spatially and temporarily, the amount of Chl-a can significantly vary with the abiotic processes, sediment composition and rheology (Le Hir et al., 2007; Perkins et al., 2003) and so it becomes difficult to find a proxy for the sediment stability dependence upon MPB. On the other hand, remote sensing can be used as a synoptic method to quantify the intertidal flat stability (Hakvoort et al., 1998; Paterson et al., 1998). The sensitivity analysis of the bio-cohesivity parameter (α), suggests that the final channel morphology and the sediment composition of the substrate is strongly affected by small changes of this parameter that correlate biofilm biomass with sediment stability (Figure 7). To fully reproduce, numerically, the influence of biological cohesion in different habitats is still a challenge due to the complexity of intertidal systems; benthic communities and the differences in physical and biological processes across sediment grain sizes, and a more detailed parametrization of the effect of the MPB community is required to properly describe these environments (Hope et al. 2020).

The results presented herein demonstrate that biophysical scale-dependent feedbacks are crucial in regulating the substrate and the spatial self-organization of intertidal ecosystems. This process is fundamental not only for the development of present channels, but also dating ancient deposits, for example during the Precambrian when biofilms were the predominant benthic ecosystem (van de Vijsel et al., 2020). The influence of biological processes on morphodynamic evolution of these aquatic environments, and their impact on the geological record, remains largely unquantified. Significant knowledge gaps remain on how small-scale biological activity can impact large-scale cohesive sediment dynamics and overall landscape evolution. Earth's geological record provides evidence for long-lived sediment biostabilisation (e.g. microbialites; Burne and Moore, 1987; stromatolites; Hohl et al., 2021). These deposits are formed by microbial communities form sticky microbial mats, excreting similar glue-like EPS that interact with the flow and the sediment transport processes. The similarities of





these deposits with present-day biofilms (in terms of layering, morphological pattern and formation mechanisms) serve as an equivalent to Precambrian and Phanerozoic microbialites (Noffke, 2008; Noffke et al., 2013). The abundance of well-preserved layers of microbialite fossils in the geological record make them especially valuable for paleoenvironment reconstruction. In fact, these primitive biostabilising organisms can play a major role in sedimentary landscape development,

for instance stimulating self-organization of mudflats (Weerman et al., 2010), with sediment layering promoted by their seasonality, which enhances the regularly patterned development of bedforms. Further development of the model is required to account for the long-term effect of sticky microbial biofilms on the substrate and its effect on the landscape development.

While this study provides a sensitivity analysis of the biofilm model parameters, several assumptions and simplifications of the complexity of the biomorphology of these environments have been made. Resuspension of MPB in the water column in

highly productive ecosystems will promote the establishment of surface biofilms in adjacent habitats. In estuarine and intertidal environments, the MPB, macrophytes and zoobenthos are rarely uniformly distributed, rather they are found in patches with a highly variability, though typically rather small scale. Patchiness adds to lack of uniformity related also to the grain size characteristics at the bed, but this study does not account for patchiness and non-uniformly spatially distributed biofilm. Empirical observation shows that self-organization due to scale-dependent feedback between biological and

geomorphological processes locally improve living conditions (Weerman et al., 2010). The regularly spaced patterns are characterized by elevated hummocks with high diatom biomass and higher sediment erosion thresholds, alternated by water-filled hollows with lower biomass and lower stability. The presence of these patterns on the bedforms, further regulate the pattern development as the hollows are less stable than the hummocks.

Cohesivity due to the presence of microbial EPS inhibits sediment transport, bedform size and development (Parsons et al.,

2016) and it can significantly affect large scale geomorphology; changing sedimentary habitats. This can hinder the formation of sediment bedforms, which changes the relationship between hydrodynamics and bottom shear stress (Malarkey et al., 2015; Parsons et al., 2016). Similarly, macrofaunal activity can increase the bottom roughness and surface heterogeneity that alters the benthic boundary layer (Borsje et al., 2009; Coco et al., 2006; Brückner et al. 2021), and the interactions between stabilizing microbes and fauna can add further complexity. For example, infauna excrete essential

nutrients such as nitrogen which stimulates the growth of MPB and as a consequence the bed stability (Murray et al., 2014); with the presence of burrows, mounds and tube mats increasing the surface area of sediment, and creating a patchy distribution of nutrients on the substrate enhancing the spatial complexity of biofilm distribution across the sediment. This variability facilitates new habitat formation and the development of environments favourable for larger organisms and this positive effect on microbial organisms can neutralise the destabilizing action of surface dwellers or grazing effects (Hope et

al., 2019). The presence of grazers and the abundance of nutrients can work differently at different spatial and temporal scales, and this often creates complex interaction that are difficult quantify (Posey et al., 1999). On intertidal flats, spatial self-organisation of microbes observed during early spring months can be destroyed as the season progresses. This shift towards a more homogenous surface is attributed to the presence of herbivores, bioturbation activity and the increase in grazing activity as the season progresses (Weerman et al., 2010; Weerman et al., 2011a, 2011b). Going forward, sediment



biostabilisation modelling should take into account the added effects of grazing and bioturbation processes as these affect the presence and impact of microbial biofilms and surficial sediment properties (Cozzoli et al., 2019; Brückner et al., 2021).

Lastly, physico-chemical parameters (salinity, pH) play a key role in these environments. Field studies have shown that biostabilisation, can be modulated by of the availability of ions that aid binding, with the stabilization potential in freshwater often significantly lower than marine environments (Spears et al., 2008).

**5. Conclusion**

The study presented here has provided a novel insight into the morphodynamic evolution of intertidal channels. It includes the effects of surface biofilms and accounts for the effect of seasonality and temperature changes on biostabilisation potential. The 1D biostabilisation shallow water model was implemented under different hydrodynamic conditions to investigate different climate scenarios and understand which are the biofilm development parameters that influence the final

channel morphology.

The model can also be utilised to investigate the bed and deposit evolution of a tidal dominated channel, starting from a horizontal bed until it reaches equilibrium. The output suggests that high hydrodynamic disturbances play a fundamental role in shaping the channel equilibrium profile, by creating carpet-like erosion, which removes the biofilm layer and exposes the clean sediment underneath. Low hydrodynamic forces (e.g. supratidal area) allow the steady development of biofilm, and the

consequent biostabilisation can inhibit sediment mobility, strongly controlling the final bed profile. The frequency and intensity of the hydrodynamic disturbances, therefore regulates the growth and stability of the biofilm.

Changes in the annual sediment temperature profile (for instance due to climate change), or of the biofilm maximum growth factor (regulated by e.g. nutrient availability) strongly influences the amount of surface biofilm, and as a consequence the bed profile and stratigraphy. Increasing and decreasing the sediment temperature from the optimal temperature for

photosynthesis, both result in a less stable and less developed biofilm, and as a consequence the bed is more mobile.

It is concluded that hydrodynamic forces play a decisive role in shaping the geometry of the channel also in the uniform presence of surface biofilm but the stratigraphy of the deposit is significantly affected by the biofilm conditions, with coarse sediment trapped when there is a stable biofilm present.




## Appendix: Model Validation and sensitivity analysis


The one dimensional shallow water equations modified for partially dry areas are solved simultaneously using the explicit, second-order accurate in space and time predictor-corrector MacCormack scheme (Chaudhry, 2008; Viparelli et al., 2019). The numerical model is implemented on tide dominated horizontal channel subject to tidal fluctuation at the ocean boundary, which result in erosion in the ocean part and a landward migrating shoal, depositing and forming a beach until it reaches

equilibrium conditions. The domain is divided into N-cells of width $\Delta x$, set equal to 0.5 m to have enough spatial resolution. The bed and water surface elevation with respect to the datum are denoted by $\eta(i)$ and $\xi(i)$ respectively.

An impermeable wall is assumed at landward boundary ($Q|_{x=0} = 0$, $Q_b|_{x=0} = 0$). An open ocean or tidal basin is assumed at the ocean boundary (x = L) with amplitude $\alpha_t$ and periodicity $\omega_t$, from where tides propagate into the domain:

$$\xi_d = \xi_0 + \alpha_t \cos(2\pi t/\omega_t), \tag{A1}$$

Extra points are added at the land and ocean boundaries of the domain to compute the predictor and corrector terms respectively, zero gradient for discharge and water surface elevation is assumed at the land boundary (x=0) while at the ocean boundary the flow rate and the water surface elevation are set equal to the value at (x=L) (Viparelli et al., 2019).

The final numerically modelled bed profile after 2,000 tidal cycles shows good agreement with the temporal evolution of the cross-sectional averaged bed profile Tambroni et al. (2005) obtained from laboratory investigation of the process whereby an

equilibrium morphology is established in a tidal system consisting of an erodible channel connected through an inlet to a tidal sea (Figure A1). The bed profile generated from the numerical model show weaker concavity of the bed profile, resulting in better match with the theoretical predictions suggested by Seminara et al. (2010). Seminara et al. (2010) proposed two theoretical predictions for tidal dominated channels, assuming Chezy coefficient as constant ($C_{constant} = 12$) or as function of the outer bottom profile at equilibrium ($D_0$; $C_{variable} = C_0 D_0^{1/6}$) (dashed lines in Figure A1). The numerically

simulated channel slightly underestimates the bed elevation at the entrance at the landward boundary (Figure A1).

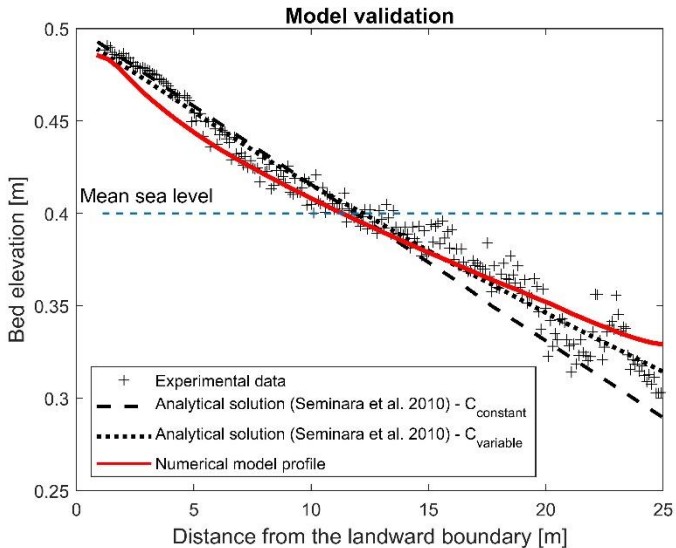



**Figure A1: The experimental bed profile (grey symbols) observed by Tambroni et al. (2005) after 2000 tidal cycles in a straight, tidal channel with constant width, and the theoretical predictions (two dashed lines) resulting from equations suggested by Seminara et al. (2010), computed with a constant and variable Chezy flow conductance, are compared with the modelled bed profile (red line)**

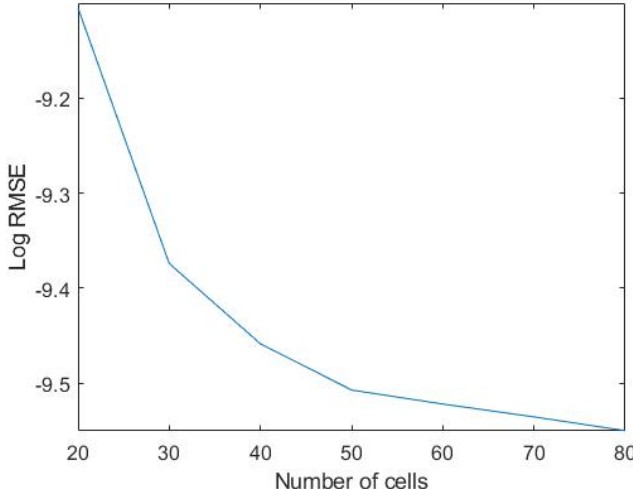

**Figure A2: Logarithmic RMSE from the comparison between the model run with different number of computational cells (x axes) and the analytical solution computed with a variable Chezy flow conductance, Seminara et al. (2010)**

Grid-sensitivity analysis has been performed by investigating different range of computational grid points in the streamwise direction. Increasing the grid resolution did not show any significant effect on the results (Figure A2).

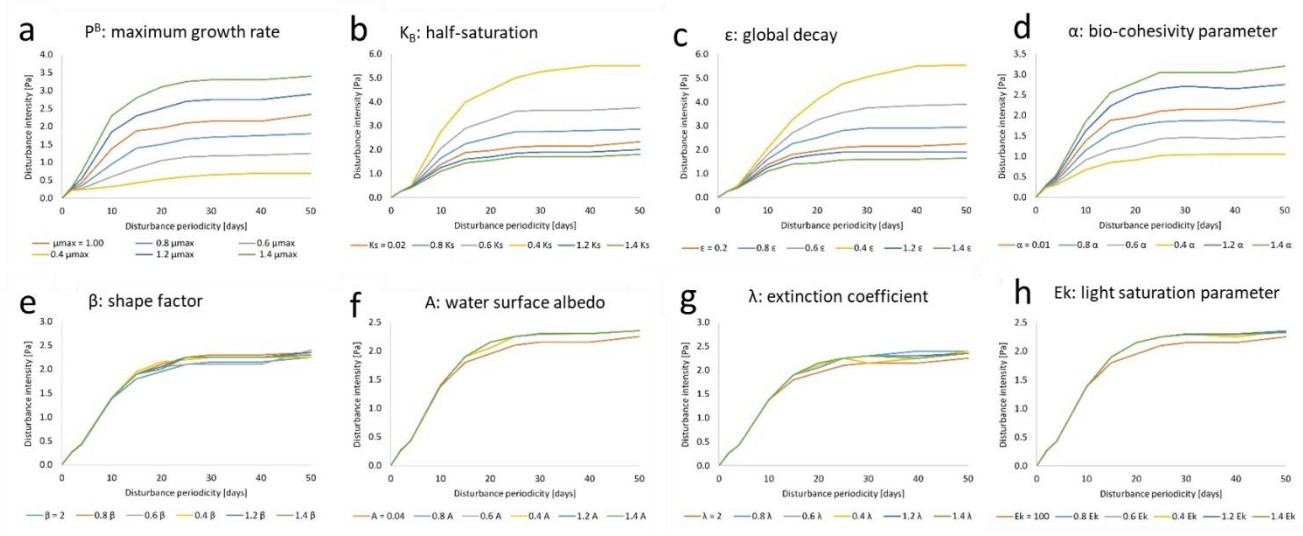

**Figure A3: Sensitivity analysis of the model parameters on the determination of the equilibrium configuration.**





**List of symbols**

| A | Water surface albedo
| $a_r$ | Characteristic length scale of the bed irregularities
| $A_i$ | Cross sectional area averaged over bed irregularities (W $\xi$)
| B | Biofilm biomass
| $B_{min}$ | Background biofilm
| $c_{0i}$ | Near-bed concentration of suspended sediment in the generic grain size range averaged over turbulence
| $C_f$ | Friction coefficient
| $c_i$ | Volumetric sediment concentration
| $D_{50}$ | Median diameter of the bed material
| $D_g$ | Geometric mean sediment grain size
| $D_i$ | Characteristic diameter
| E | Catastrophic erosion
| $E_i$ | Grain size specific entrainment rate under equilibrium of suspension
| $E_k$ | Light saturation parameter
| $E_T$ | Entrainment rate per unit bed summed over all the grain-sizes
| $E_i$ | Entrainment rate per unit bed for each grain-size i
| $F_H$ | Wet fraction of the channel bed
| g | Acceleration of gravity
| $H_{res}$ | Light availability
| $K_B$ | Half-saturation constant for biofilm growth
| L | Channel length
| N | Number of computational nodes
| $p_i$ | fraction of sediment in each grain-size range
| $P^B$ | Effective maximum growth rate for biofilm
| $P_{max}$ | Maximum growth rate for biofilm
| $P^B_{max}$ | Biofilm growth rate under light saturation conditions
| Q | Flow discharge ($A_c$ U)
| $Q_b$ | Total material load as the sum of the contribution of bedload and suspended load summed over all the grain sizes
| $Q_{b,bi}$ | Total volumetric bed material load as the contribution of bedload, for the generic grain size i
| $Q_{b,si}$ | Total volumetric bed material load as the contribution of suspended load, for the generic grain size i
| R | Submerged specific gravity of the bed material
| $R_H$ | Hydraulic radius($A_c$ / $\chi$)



| | | |
|---|---|---|
| | $R_{sun}$ | Solar irradiance reaching the water surface |
| | $S_f$ | Friction slope |
| | $t$ | Temporal coordinate |
| 740 | $t_i$ | Time detachment due to high hydrodynamic forces |
| | $\tau_b$ | Average bed shear stress |
| | $T_{s0}$ | Surface sediment temperature |
| | $T_{max}$ | Maximum temperature for photosynthesis |
| | $T_{opt}$ | Optimal temperature for photosynthesis |
| 745 | $U$ | Flow velocity |
| | $u_{*c}$ | Critical shear velocity |
| | $u_{*s}$ | Shear velocity due to skin friction |
| | $v_{si}$ | Fall velocity in each grain-size range |
| | $W$ | Channel width |
| 750 | $X$ | Longitudinal coordinate |
| | $Y$ | Effective flow depth |
| | $\alpha$ | Bio-cohesivity paramenter |
| | $\alpha_t$ | Tidal amplitude |
| | $\beta$ | Shape parameter |
| 755 | $\varepsilon$ | Global decay |
| | $\delta$ | Dirac function |
| | $\eta$ | Bed profile |
| | $\eta_o$ | Initial bed elevation |
| | $\lambda$ | Extinction coefficient |
| 760 | $\xi$ | Mean water surface elevation |
| | $\rho_s$ | Density of the sediment |
| | $\tau_b$ | Bed shear stress |
| | $\tau_{bc}$ | Critical shear stress for erosion |
| | $\tau_{bc,0}$ | Clean sediment critical shear stress |
| 765 | $\tau_{bs}$ | Bed shear stress due to skin friction |
| | $\tau_i^*$ | Grain size specific Shields number |
| | $\tau_{bc,i}^*$ | Grain size specific reference Shields number for significant bedload transport |
| | $\chi$ | Wetted perimeter |
| | $\omega_t$ | Tidal period |
| 770 | $\zeta$ | $= z/b$; dimensionless upward normal coordinate |





**Author contributions**

EB developed the model, EB and DRP designed the numerical simulations; EB, RMD and JAH contributed to interpretation of the results. EB drafted the manuscript, all authors contributed in editing the manuscript.

**Competing interests**

The authors declare that they have no conflict of interest.





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
