# Peer review of "Effect of hydro-climate variation on biofilm dynamic and impact in intertidal environment"

_EGUsphere, 2022_

## Author Comment (AC1)

**Review 1**

We thank the reviewer for taking the time to review our paper and provide responses below. These have significantly improved the clarity and presentation of our manuscript.
A point-by-point description of how we (the authors) addressed suggestions is presented below. Comments are in black, our responses are in blue and modified text in the manuscript is in green. When we use line numbers, we refer to the resubmitted paper (new lines) with no track changes.

**Major Comment 1:**
*One concern is that the biofilm-dependent erodibility model and the model describing the biofilm biomass dynamics (i.e., all of section 2.3) does not seem to have any backing by observations or data other than the reference to Friend et al. (2002) in line 245. This reference identifies the temperature controls on the growth of MPB. Nevertheless, there is no evidence presented that the behaviors displayed in Figure 2 are representative of reality. They may very well be, but the evidence is not explicitly provided, calling into question what the sensitivity analysis results mean (e.g., if the base parameters in the model are not representative, what does the sensitivity analysis teach us?). That does not mean, in my opinion that the analysis is invalid, but I do think that the authors have an opportunity to more clearly and convincingly address the validity of the models presented in section 2.3. This will enhance the reader's confidence in the results presented in the following section.*

We thank the reviewer for these comments. We acknowledge we could have been clearer concerning how the range of values and parameters used within the biofilm model were determined and selected.
Reference model value for the biofilm development model (table 2) are based on steady state equilibrium of biofilm biomass in intertidal environment, which is assumed equal to about 200 mg Chl-a/m$^2$ (figure 2a) in temperate areas. These values were calculated from those reported in literature from *in-situ* field measurements (Mariotti and Fagherazzi, 2012; Le Hir et al., 2007). The surface biofilm biomass is modulated by the annual variation of the sediment temperature according to the sediment temperature model proposed by Pivato et al. (2019), which is based on data from the Venice Lagoon (figure 2b). Figure 2c provides an example where the periodic removal of biofilm and the consequent change in critical shear stress, occur – highlighting the impacts via a sensitivity analysis.
The first set of sensitivity analysis look at the impact of biofilm parameters on biofilm stability (Figure 3), by changing biofilm function parameters (such as biofilm maximum grow rate, half-saturation constant, global decay, bio-cohesivity parameter, shape factor, water surface albedo, extinction coefficient and light saturation parameter) around their reference value (0.6 ÷1.4 the reference value). The sensitivity analysis of biofilm model parameters (temperature, nutrients, bio-cohesivity parameter) on channel development substrate, has been performed by changing these parameters within the range identified from the literature (Mariotti and Fagherazzi, 2012; Pivato et al., 2019). The results are therefore seeking to cover the range of behaviors related to this range of parameters, giving insights on the biofilm development at different environmental conditions.
We have updated the text of the manuscript to more clearly identify how and where the range in parameters we have used have emerged from, identifying more clearly the sources of literature used (**lines 237 to 242, lines 254 to 256**, **a column with the range values for the biofilm parameters is added to Table 2 and its caption has been updated, lines 294 to 295**).

**Lines 237-243:**
The reference values for the parameters of the biofilm growth function (Table 2) are based on field observations, assuming that in equilibrium conditions the surface biofilm biomass is equal to 200 mg Chl-a/m2 (Mariotti and Fagherazzi, 2012; Le Hir et al., 2007), which is a commonly

found in intertidal environments in temperate areas (Mariotti and Fagherazzi, 2012, Le Hir et al., 2007). At the initial stages, the growth of undisturbed biofilm is approximately exponential then, as the saturation begins, it slows to linear until it reaches maturity when the growth stops and the amount of biofilm on the bed surface remain constant for the entire duration of the simulation, reaching asymptotically an equilibrium condition (Figure 2a).

**Lines 254-256:**
The sediment temperature model implemented in this study account for the effect of seasonality, and it is based on sediment temperature parameters based on temperate environments as proposed by Guarini et al. (2000), Pratt et al. (2014), Pivato et al. (2019) (Table 2).

**Line 294-295:**
The range of values found in literature and the reference values selected here are reported in Table 2.

**Major Comment 2:**
*It is unclear to this reviewer the ranges of the sensitivity parameters being tested. I see in lines 290-296 that a number of different model parameters are being tested, but I have no clear understanding of how they will be modulated, how many model runs will be performed, if parameters will be tested in isolation or a scenario format. Essentially, I have no idea what to expect to see in the results of than some form of model results. The manuscript would benefit from a clear description of the testing scheme. It's fairly clear after reading the results, but comprehension would have increased with a table of tabled parameters in the methods or perhaps pulling Fig.A3 into the main text to show the tested parameters.*

We have updated the manuscript to include this information for the reader. We have updated **table 2** with a column providing the range of values from literature for the model parameters and a column with the reference values used in the simulations. A clear articulation in the text of the series and number of simulations undertaken (**see lines 298 to 315**). Figure A3 has been moved to the main text and a summary table of the simulations has been added (**Table 3**).

**Lines 298-315:**
By changing the biofilm model parameters within the range found in literature, this study investigates the sensitivity of the key biofilm model parameters on the morphological evolution of an intertidal channel (Table 3).
Firstly, it is presented a sensitivity analysis of the biofilm stability under different hydrodynamic disturbances characterized by periodicity (T) and intensity (τ_0, shear stress). The sensitivity analysis is performed for all the model parameters and within the range of values suggested in literature (0.6÷1.4 times the reference value suggested in literature, Table 2), by systematically changing the periodicity and intensity of the hydrodynamic disturbances to evaluate under which conditions the biofilm is stable. Then, with the objective to test the biofilm stability modulated by the effect of seasonality, it is performed a sensitivity analysis of the maximum and minimum values of the parameters that are strongly affected by seasonality, like the biofilm grow rate (PB) and the sediment temperature (T + 5 °C, T – 5 °C).
A second set of sensitivity analysis test aim to understand the bio-modulation of channel morphodynamics evolution for an idealized channel characterized by semidiurnal tidal. The results show the comparison between the impact on biostabilization for uniform and spatially varied surface biofilm under the effect of carpet-like erosion, which is regulated by the periodic hydrodynamic disturbances changes in the water level at the seaward boundary due to the tidal forces, and seasonality. In the case of combined effect of these conditions (seasonality and carpet-like erosion), it has been investigated how channel morphology changes with the variation of i) biofilm growth rate; ii) sediment temperature; and iii) sediment bio-cohesivity.

The numerical simulations have been performed in the absence of an imposed input of sand from the ocean, and without riverine water and sediment at the landward boundary. It has been assumed an idealized 25 m long channel with constant width equal to 30 cm.

**Major Comment 3:**
*For the "strong" and "weak" disturbances, it would be useful to provide descriptions of what types of events these are representing (storms? Regular tides? Especially high tides?) to go alongside the shear stress values presented. It's also unclear to me whether the intervals of disturbance chosen are representative of anything "real". What does 15/5 day intervals represent? Is it an arbitrary selection? Or is this a spring and neap cycle? I understood that the tidal forcing was only semidiurnal.*

We have added these details into the text to make this clearer to the readers (**Lines 319 to 321, Lines 338 to 342**). The intensity and the periodicity of the hydrodynamic disturbances simulated in the analysis presented in Figure 3 are equal to those studied by Mariotti and Fagherazzi (2012, Figure 2) which have been selected randomly to investigate the temporal modulation of the disturbance and their impact on biofilm development, especially in the case that disturbances systematically destroy biofilm.

**Lines 319-321:**
The sensitivity analysis of the biofilm model parameters (Equation 2-3-4-5-6) have been investigated by systematically changing the intensity and the periodicity of the disturbances, to find the hydrodynamic conditions at which the status of the biofilm change from stable to detached (Figure 3)

**Lines 338 to 342:**
The sensitivity analysis is carried out for a year-long cycle, the intensity and periodicity of the hydrodynamic conditions are selected from the previous analysis according with what has been observed by Mariotti and Fagherazzi (2012). High-intensity and low disturbance periodicity events (case 1 in Figure 4, T = 15 days, $\tau_0$ = 1.5 Pa) are assumed to allow the growth of biofilm under reference values for the biofilm model parameters, while under frequent and weak disturbances (case 2 in Figure 4, T = 5 days, $\tau_0$ = 0.5 Pa) the biofilm is not fully established on the bed.

**Lines 536-544:**
The simulations presented in Figure 4 assumes periodic disturbances to investigate which is the effect of changes of biofilm model parameter on biofilm establishment and growth. The frequency and intensity of storms is likely to increase in the future due to climate change, and the resulting drastic morphological changes on tidal flats can occur over short durations. This will affect biofilm evolution and establishment and therefore the degree of biological stabilization that occurs. Storms can induce strong wave activities, elevate water levels and cause severe erosion of tidal flats due to enhanced bed shear stress and carpet-like erosion of surface biofilm. The associated high suspended sediment concentrations and long inundation period increase the turbidity on the water column and inhibit photosynthesis. The model presented here can be easily adapted to account for the seasonal variability in storms by incorporating the combined hydrodynamic effects of occasional storms and periodic tidal forces.

**Major Comment 4:**
*How are the hydrodynamic disturbances imposed on the model? Are these done by drastically increasing water level at the seaward boundary or is there a momentum component? Or is there wind in the model? Over a small domain (25m), I would doubt it. I think this confusion could be rectified by addressing the previous comment and then briefly describing how the disturbances are implemented.*

Changes in the water level at the seaward boundary (e.g. due to the tidal forces) were set to create periodic hydrodynamic disturbances. The intensity of the correlated shear stresses ($\tau$) computed at the bed were then used to determine if the surface biofilm will be removed or not (if $\tau > \tau_{bc}$, sediment). While for the analysis presented in Figure 4 the hydrodynamic disturbances are set periodically and they all have the same intensity to investigate how this affect the establishment of surface biofilm. Again, we have modified the description within the text (**lines 307 to 310, lines 392 to 394**) to improve clarity.

**Lines 176-178:**
Shallow water equations model (Chaudhry, 2008) are used to describe temporal and 1D spatial variation of idealized tidally-dominated channel reaches (Figure 1). For example, these might include submerged river-valley coastlines, lagoons or shorelines protected by natural or artificial barrier islands where surface waves may be assumed negligible.

**Line 308-310:**
The results show the comparison between the impact on biostabilization for uniform and spatially varied surface biofilm under the effect of carpet-like erosion, which is regulated by the periodic hydrodynamic disturbances changes in the water level at the seaward boundary due to the tidal forces, and seasonality.

**Line 392-394:**
The effect of carpet-like erosion is modulated by changes in the water level at the seaward boundary due to the tidal forces creating periodic hydrodynamic disturbances.

**Minor Comments:**

**Lines 11-14:** There is something wrong with grammar of this sentence. The list of metrics ( or perhaps those are the tuning parameters for the sensitivity analysis – it's not clear) isn't explained. I think I get the point but the sentence needs to be restructured for clarity.

**Lines 11-14** are changed to:

This study investigates the effect of a range of environmental and biological conditions on biofilm growth, and their feedback on the morphological evolution of the entire intertidal channel. By carrying out a sensitivity analysis of the bio-morphodynamic model, parameters like i) hydrodynamic disturbances; ii) seasonality; iii) biofilm growth rate; iv) temperature variation; and v) bio-cohesivity of the sediment, are systematically changed.

**Line 18 & 317 & 465 & 595:** Can't say "on the other hand" without a previous sentence saying "on one hand." It's a pet peeve of mine :)

'On the other hand' has been substituted with 'while' or deleted where appropriate.

**Line 22-23:** rephrase to "…predict estuary development and mitigate coastal erosion"

The sentence has been modified accordingly (**lines 21-22**)

It is concluded that inclusion of biocohesion in morphodynamic models is essential to predict estuary development and mitigate coastal erosion.

**Line 40-48:** I'm wondering if the authors could add a quick comment somewhere in here about the global ubiquity of MPD and EPS on intertidal and subtidal channels. Is it present throughout the world? If so, how prevalent is it?

The text has been updated including this overview of biofilm biofilm global ubiquity.

**Lines 43-47:**
Biofilms composed of MPB and EPS are ubiquitous in aquatic sediments (sand and mud) from shallow fluvial systems to continental shelves still within the photic zone (Cahoon 1999), even under physical disturbance from flow (Hope et al., 2020; Pinckney et al., 2018). While prevalence and patchiness can be greater on intertidal muddy flats, biofilm distribution in sandier intertidal and subtidal channels can be more homogenous as seen in the Western Scheldt (Daggers et al., 2020).

Grammar in Section 2.3 and beyond: I'm noticing several sentence fragments, missing punctuation, issues with parentheses, and noun-verb plurality disagreements. I can still understand the text but it's distracting. (Examples include lines 247-249, line 245, line 221, line 249, and further along with grammatical errors in line 296, line 299, lines 347-352, among others). I stopped noting them after about line 350, but to be increasing in frequency as the manuscript goes on.

We have now proofread through the manuscript for grammatical mistakes and flow.

**Lines 347-350:** A reference to Fig A3 is appropriate here. It took me a while to figure out what this sentence meant until I search through the appendix.

Figure A3 has been included in the main text (now Figure 3)

Figure 4: The inset figures in c, g , and k are extremely small. I would also suggest using the figure itself to highlight assumptions for each row. Perhaps this means labeling them in the figure by highlighting the rows, or similar (this is already done for the columns). With a full page figure, I keep having to scroll back and forth between the figure and the description to figure out what I'm looking at.

Figure 4 (now Figure 5) has been replotted and made clearer. The figure shows bed evolution for the reference case (clean sediment) and all the simulated biofilm scenarios, with a panel showing the biofilm annual pattern. The analogue figure representing the grain size distribution of the substrate have been added to the appendix (Figure A3).

**Line 449**: I know it's not used in the simulations shown in Figure 6, but I'm still very unclear about what "high hydrodynamic forces" means from primarily the physical perspective ( I understand the quantitative shear stress side).

As it has been shown in Figure 5, carpet-like erosion can have quite a strong effect on the final channel morphology. Since this section focuses on the effect of seasonality and changes in temperature, it has been decided to neglect the effect of carpet-like erosion (turn it off in the model). This means that the results presented here can be associated with areas of low hydrodynamic activities (waves, tides) and therefore the biofilm development is dominated by the effect of temperature and seasonality.
To ensure clarity, the sentence has been rephrased as below (**lines 461-463**):

**Lines 461-463**:
The simulations presented here focus on the effect of seasonality and changes in temperature, therefore the effect of carpet-like erosion is neglected.

**Lines 473-474:** "This result in a more stable bed, morphological changes occur in a longer timeframe and, by the end of the simulation, it does not reach an equilibrium condition"…I don't understand this argument. Why would it reach and equilibrium state along the same timeline as the simulation not including the biomass growth and biostability? It seems they are two entirely different simulations that can certainly be compared, but I don't think it's a

foregone conclusion that they should reach the same steady state condition simultaneously. However, it does appear in Figure 6c that it is approaching some type of steady state, albeit potentially significantly more slowly than in Figures 6b or d. Were the simulations extended out beyond ~10^4 tidal cycles to verify it does not reach an equilibrium state?

We agree, we are not expecting the intertidal channel to reach the same final equilibrium profile, under all the different conditions simulated. All the simulations were run to the same duration (30,000 tidal cycle ~ 40 years) for comparisons to be made, we did not extend beyond this point as we feel that this is an adequate timescale in which to compare differences. We have rephrased this sentence to clarify.

**Line 446-447:**
Under this condition, the bed after 30,000 tidal cycles still dynamic both at the landward and seaward boundary.

**Line 529-530:** I'm wondering what the justification is for extending the results presented here to estuarine stability. The results presented only test the effects of disturbances and biomass growth on tidal channel morphology in one dimension. This cannot account for the myriad factors contributing to whole estuarine morphology. In fact, the model presented here does not even model an estuary at all, but a single tidal channel. The connection seems tenuous. I think the authors should either omit this point of discussion or stringently justify the extension to whole-estuarine morphology.

The sentence has been modified

**Lines 552-555**:
Consequently, an increasing extent and thickness of mud cover might lead to a stabilization of large-scale estuarine morphology. Although not directly modelled in this study, as our findings suggest that the sediment bed would become 'muddier' as biostabilisation is increased these changes may influence wider estuarine morphology as channels are stabilised, attract more mud and influence the evolution of channel morphology.

**Line 543:** The line here states the model is validated for temperate areas. I don't see any comparison to data. Was it validated in another paper? I looked throughout the manuscript and didn't find any mention of this. This is the only mention of validation is in regard to the morphodynamic model along (Appendix Figure A1) This bring me back to the Major comment #1.

Appendix 1 shows the validation of the only morphodynamic model. This study presents a sensitivity analysis of the bio-morphodynamic model by changing the model parameters within the range proposed in literature, to investigate which is the effect on intertidal channel morphology. The reference parameters adopted from literature are based on field observations at temperate regions. The simulations presented aim to investigate how changes in those parameters due to climate change can affect the morphology and the substrate of intertidal channels (also see comment 1 for more details). We have also altered the text (see below) to clarify this for the reader.

Appendix Figure A3: This is not discussed at all in the appendix text.

The figure has been moved to the main text (now Figure 3) and is referred to in section 3.1 (e.g. **Lines 333, 346**)

Line 630: This is a sentence fragment or something.

The sentence has been revised:

**Lines 736-742:**
Similarly, macrofaunal activity can increase the bottom roughness and surface heterogeneity that alters the benthic boundary layer (Borsje et al., 2009; Coco et al., 2006; Brückner et al. 2021), and) enhancing the complexity of the interactions between stabilizing microbes and fauna can add further complexity.macrofauna. For example, infauna excrete essential nutrients such as nitrogen which stimulates the growth of MPB and as a consequence thetherefore bed stability (Murray et al., 2014); with the presence of). Further, their burrows, mounds and tube mats increasingincreases the surface area of sediment, and creating a patchy distribution of nutrients on the substrate enhancing the spatial complexity of biofilm distribution

**Lines 577-649:** I think this section can be significantly shortened. There is a lot of addition literature review here with only a few lines relating the material studied in this paper to known processes/effects published in the literature. I encourage the authors to trim this down and focus more on contextualizing their results within the current understanding of biostabilization and the state of modeling such processes (as alluded to in 616-617).

The discussion session has been reviewed accordingly.

**Review 2**

Thank you very much for your thorough and thoughtful comments. We believe that in addressing these comments, the manuscript has improved significantly.
A point-by-point description of how we have addressed suggestions is presented below. Comments are in black, our responses are in blue and modified text in the manuscript is in green. When we use line numbers, we refer to the resubmitted paper with no track changes.

**Comments:**

What about seasonal variability in erosive events? I don't think it is necessary to include new model runs of this in the manuscript (you already did a lot of model runs!), but I think it would be worth discussing the impacts of a system that has seasonal variability in storms – where the two different disturbance scenarios you describe may both exist within a year but at different times.

The manuscript has been updated and modified based on suggestions (**lines 532-544**). Special attention has been given to how high-energy activities can cause severe erosion of tidal flats due to enhanced bed shear stress and carpet-like erosion of surface biofilm, as well as the inhibition of photosynthesis due to the high inundation period and the high sediment in suspension that increase turbidity in the water column.

**Lines 532-544:**
Moreover, high bed shear stresses due to hydrodynamic forces (tides) can cause a general delay in biofilm formation and biostabilisation (Figure 4) and a significant decrease of the biofilm stability (Schmidt et al. 2018). This study does not incorporate the combine hydrodynamic effect on surface biofilm mass of occasional storms and periodic tidal forces. Morphology and sedimentary processes on tidal flats can be strongly affected by storms and associated high-energy activities over a short time. The simulations presented in Figure 4 assumes periodic disturbances to investigate which is the effect of changes of biofilm model parameter on biofilm establishment and growth. The frequency and intensity of storms is likely to increase in the future due to climate change, and the resulting drastic morphological changes on tidal flats can occur over short durations. This will affect biofilm evolution and establishment and therefore the degree of biological stabilization that occurs. Storms can induce strong wave activities, elevate water levels and cause severe erosion of tidal flats due to enhanced bed shear stress and carpet-like erosion of surface biofilm. The associated high suspended sediment concentrations and long inundation period increase the turbidity on the water column and inhibit photosynthesis. The model presented here can be easily adapted to account for the seasonal variability in storms by incorporating the combined hydrodynamic effects of occasional storms and periodic tidal forces.

The authors do a good job describing the relationship between biofilm biomass and chla, but neglect to really mention EPS production directly. I would recommend to add a paragraph to the introduction that discusses the relationship between EPS and erodibility (which is much stronger than the chl a – erodibility relationship, and has a stronger process-based explanation), and why you chose to model chl a instead of EPS production. There are some particularly interesting studies that point out that EPS production can be linked to other factors (like nutrient availability, biofilm stress, see Ruddy et al. 1998, Smith and Underwood 2000, Underwood 2002, Orvain 2003, Orvain 2014, Hubas 2018) that may effect erodibility. While I think modeling chl a/biofilm biomass as a proxy for erodibility is fine (and standard practice), I think mentioning these limitations is important.

Thank you, we have now amended the text to introduce EPS earlier in the introduction, discuss it's relationship with erodibility and other factors such as

**Lines 48-54:**
It has been shown that secreted EPS is crucial in the adhesion/cohesion of the substratum and sediment particles, and it can act as a protective layer at the bed surface reducing the bed roughness, influencing significantly the erosion and deposition of sediment particles by raising the sediment erosion threshold due to cohesion (Tolhurst et al. 2002, Tolhurst et al. 2006, Tolhurst et al. 2009, Paterson et al., 2018, Hope et al. 2020). This promotes the sedimentation of fine-grained particles and subsequently stimulates biofilm growth (Weerman et al., 2010) as nutrient are supplied to the bed. Microbial production of EPS is influenced not only by nutrient availability, but can be stimulated with exposure to contaminants such as heavy metals and nanoparticles (Ruddy et al. 1998; Lubarksy et al., 2010; Joshi et al., 2012).

**Line 119-125:**
Numerous studies in marine intertidal environments show a positive correlation between sediment stability in terms of critical shear stress for erosion ($\tau\_bc$) and EPS components of biofilm. Although it is EPS that stabilises the bed, not the MPB per se, chlorophyll-a (Chl-a), a proxy of living MPB biomass, provides a good approximation of biostabilisation potential (Defew et al., 2002; Paterson et al., 2000; Riethmuller et al., 2000, Haro et al., 2022). Chl-a is often the preferred measurement, due to its ecological significance and the fact that it is easy to evaluate (both in the field and by optical remote sensing) (Andersen, 2001; Le Hir et al., 2007), but Chl-a – stability relationships can often be weak, emphasising the complexity of this phenomenon and that important interactions are being missed.

Is there residual EPS in the sediments? Once there is erosion, should the sediment really return to an abiotic state? I think this is an assumption that affects your results (also the results of Mariotti and Fagherazzi 2012, Pivato et al. 2019, and others). I think this merits some discussion in the text. There is some evidence that even after erosive events, some remnants of biofilm or EPS may lead to faster biofilm establishment (Chen et al. 2019) or changes in erodibility after repeated erosion (Valentine et al. 2014).

We agree, and have introduced this concept in the introduction for the reader (Line 144).
The biofilm growth function implemented in the model accounts for the background (remaining) biofilm biomass, which considers the recolonization after biofilm removal ($B_{min}$ = 1). The background biofilm conditions are assumed in case of chronic and self-generated biofilm detachment (equation 3), and also in the case of carpet-like erosion (equation 8).
We have also modified lines 287-292 to clarify this to the reader.

When the stress on the bed exceeds the critical value for erosion ($\tau_{bc}$), the biofilm is eroded and is reduced to the background value $B_{min}$. This background value is to account for the remaining MPB cells which re-establish the biofilm (Figure 2c). However, when biofilm is removed from the bed surface as carpet-like erosion, the resistance of the bed reduces to a minimal value (Figure 2c) under the assumption of linear relationship between surface biofilm biomass and critical shear stress for erosion (Le Hir et al., 2007).

**Lines 142-144:** Even when biofilms are removed during tidal inundation, the remaining MPB community can quickly re-establish itself, depending on the prevailing conditions, with a subsequent increase in biostability, as cell numbers increase and EPS secretions once again build up (Valentine et al., 2014; Chen et al., 2019; Hope et al., 2020).

**Lines 287-292:** In the case that the stress on the bed exceed the critical value for erosion ($\tau_{bc}$), the biofilm is eroded and it is reduced to the background value $B_{min}$, which allow establishment and growth of biofilm (Figure 2c). When biofilm is removed from the bed surface as carpet-like erosion, the resistance of the bed reduces to a minimal value (Figure 2c) under

the assumption of linear relationship between surface biofilm biomass and critical shear stress for erosion (Le Hir et al., 2007). This simplified model assumes that in the case of extreme hydrodynamic events, the erosion is on the order of mm-cm which is much larger than the thickness of the biofilm thickness (μm-mm).

**Figure Comments:**

Suggestions have been implemented in the manuscript.

I think one big area for improvement in this paper is in the figures. There are a lot of figures with a lot of panels and they are hard to digest as a reader. I have some suggestions about how to improve them to make your arguments stronger! I hope they are helpful.

Generally, I think all figures that showed the evolution of the profile had too many lines which made it difficult to read the figure.

Figure 1 – In the caption, it should be "represents". Additionally H is not listed in the caption and the dots on the figure (representing grain size?) are not labeled.

Caption and figure have been modified accordingly

Figure 3 – On the x axis, you may consider labeling the months instead of using days. You reference the months in the manuscript, and it takes work for the reader to translate that to days quickly.

Suggestions have been implemented in the figure (now Figure 4)

I liked that you labeled the two columns on the figure; I think you should also label the rows more clearly (maybe a label that encompasses the first three rows that says "Growth Rate Parameter" and one that encompasses the later two rows that says "Sediment Temperature"). I think this would help with the readability of the figure. The info is in the caption, but I think it would be more effective to put more labels on the actual figure.

I typically agree that the y axis in all panels should be the same, however, it is impossible to see that there is any growth in panel a (as suggested in the text). I'm not sure what to do about this.

You also have the potential to minimize the empty space between the panels in order to make each subplot larger, which I think would look better but use up the same amount of space.

Figure 4: I would label the rows of panels, like you did with the columns. It is a lot of subfigures! I liked how you labeled the mean water surface elevation and initial bed in panel a – that was very useful. I did have some difficulty with the profiles due to the number of lines displayed on each figure, I recommend reducing the number of profiles visualized. I may also try other colorbars, as the yellow was particularly difficult to see.
This is a judgement call on your part, but I would remove the tiny subplots within panels c, g, and k. They are very small and not readable. I understand you were trying to point out what the row represented, but I think simple labels (like how you did the columns) would be more effective.

The figure 4 (now Figure 5) have been modified and split in two different figures. The one representing the changes in bed elevation under different simulated conditions is in the main text, while the one representing the final grain size distribution of the deposit has been moved in the appendix.

We are happy to discuss and change to a more suitable color palette for the figures at the proof stage.

Figure 5: Again, I recommend labeling the columns or the panels with what they represent (small, medium, and large growth rates) to make for easier reading.

Figure has been modified accordingly

Figure 6: Sorry for the repetitive comment (hopefully it is easy to do at least) – please add the labels for the temperature "treatments" for the different columns on the figure.

Figure has been modified accordingly

Figure 7: Please label the panels (or columns) with the alpha value used.

Figure has been modified accordingly

**Specific Comments:**

The manuscript has been modified according with the following specific comments.

Line 96: Should be "tidal dynamics"

Line 159-160: remove the and between tidal currents and sediment erosion.

Line 356: should be "uniformly distributed"

Line 356: I think you mean the left two columns? (instead of panels)

Line 357: You refer to the fact that some studies have found biofilms in deeper waters, which I agree! You reference some of these papers later in the paper, but I think you should add citations when you say "as it has been also suggested in the literature (cite xx)".

**Lines 383-387** have been reorganized:

One case assumes that biofilm is uniformly distributed in the entire computational domain (left two columns in **Error! Reference source not found.**), to explore the case of biofilm development also in the deepest portion of the channel. In fact, biostabilising organisms are found along the entire tidal range, from intertidal and subtidal areas, to shellfish reefs and on the continental shelf as it has been suggested in literature (Cahoon, 1999; Pinckney, 2018; van de Vijsel et al. 2020).

Line 446: Should be "biofilm growth differs"

Line 448: Should be "affect" instead of effect

Line 450: I think neglect is not the right word here. Does absent or negligible fit better?

Citations:

Chen, X., Zhang, C.K., Paterson, D.M., Townend, I.H., Jin, C., Zhou, Z., Gong, Z., and Q. Feng, 2019, The effect of cyclic variation of shear stress on non-cohesive sediment stabilization by microbial biofilms: the role of "biofilm precursors', Earth Surf. Proc. Land., 44: 1471-1481.

Hubas, C., Passarelli, C., and D.M. Paterson, 2018, Microphytobenthic Biofilms: Composition and Interactions, in Mudflat Ecology, ed. PG Beninger, Springer, 63-90.

Mariotti, G., and S. Fagherazzi, 2012, Modeling the effect of tides and waves on benthic biofilms, JGR: Biogeosciences 117: G4.

Orvain, F., Galois, R., and C. Barnard, 2003, Carbohydrate production in relation to microphytobenthic biofilms development: an integrated approach in a tidal mesocosm, Microbial Ecology45: 237-251.

Orvain, F., De Crignis, M., Guizien, K., Lefebvre, S., Mallet, C., Takahashi, E., and C. Dupuy, 2014, Tidal and seasonal effects on the consortium of bacteria, microphytobenthos and exopolymers in natural intertidal biofilms (Brouage, France), Aquatic Microbial Ecology92: 6-18.

Pivato, M., Carniello, L., Moro, I., and P. D'Odorico, 2019, On the feedback between water turbidity and microphytobenthos growth in shallow tidal environments, Earth Surf. Proc. Land. 44(5): 1192-1206.

Ruddy, G., Turley, C.M., and T.E.R. Junes, 1998, Ecological interaction and sediment transport on an intertidal mudflqat I. Evidence for a biologically mediated sediment-water interface, In: Black, K.S., Paterson, D.M., and A. Cramp (Eds.), Sedimentary Processes in the Intertidal Zone, Geological Society, London, Special Publication 139: 135-148.

Smith, D.J., and G.J.C. Underwood, 2000, The production of extracellular carbohydrates by estuarine benthic diatoms: the effects of growth phase and light and dark treatment, Journal of Phycology 36(2): 321-333.

Underwood, G.J.C., 2002, Adaptations of tropical marine microphytobenthic assemblages along a gradient of light and nutrient availability in Suva Lagoon, Fiji, European Journal of Phycology37(3): 449-462.

Valentine, K., and G. Mariotti, 2014, Repeated erosion of cohesive sediments with biofilms, Advances in Geosciences39: 9-14.

---

## Referee Report (RR1)

**Review of revision for "Effect of hydro-climate variation on biofilm dynamic and impact in intertidal environment"**

**Review date:** August 12, 2022
**Ethics:** This is my second review of the manuscript and I identify no conflicts of interest.

**Review by**: Matthew Hiatt

**Summary (unchanged from previous report):** This manuscript presents an analysis of the influences of biomass and biostabilization on 1D tidal morphodynamics. A validated hydro-morphodynamic model is presented and amended to include the effects of biostabilization on long-term (~10^4 tidal cycles) tidal channel morphology and depositional/stratigraphic patterns. The influences of hydrodynamic disturbances (frequent, infrequent, small, and large) on biofilm development are also assessed alongside the effects of temperature, biofilm development depth, and biofilm growth rates.

**Assessment (from previous report, but still applies):** Overall, the topic is of interest to readers of ESurf. The paper presents a novel combination of models on an emerging topic addressing the role of smaller scale biological processes on channel-scale geomorphology. This topic fits the journal quite well and is timely.

**Assessment of revision:** The authors have taken great care to assess and respond to my recommendations for the first draft of the manuscript. I agree with their changes and greatly appreciate the revamped subsection 2.3. With the inclusion of the additional specific details on what will be tested in the paragraphs preceding Table 3 (and the very helpful inclusion of Table 3 itself). I now think tables like Table 3 should be included in all modeling papers! I found the direction of the manuscript to be very clear. As with my first report, the results were easy to understand, but the enhancements to the methods section dramatically improved readability for me. Well done on that point.

I think the new figure 5 (a revamped version of the previously submitted Figure 4) is significantly better because it is more targeted. This improves understanding.

I appreciate the clarification on the hydrodynamic disturbances applied at the boundary in lines 310 and onward. This is helpful for contextualizing the results.

Overall, this represents a strong contribution to Earth Surface Dynamics after the careful consideration of the reviewers' comments. My recommendation is to accept in present form.